# Polymorphisms in *VEGF* Signaling Pathway Genes and Their Potential Impact on Type 2 Diabetes Mellitus and Associated Complications: A Scoping Review

**DOI:** 10.3390/biomedicines13092242

**Published:** 2025-09-11

**Authors:** Christiane Mayrhofer Grocoske de Lima, Rafaela Cirillo de Melo, Nathalia Marçallo Peixoto Souza, Paula Rothbarth Silva, Dayane Ferreira Aguiar, Luana Mota Ferreira, Waldemar Volanski, Geraldo Picheth, Fabiane Gomes de Moraes Rego, Marcel Henrique Marcondes Sari

**Affiliations:** 1Graduate Program in Pharmaceutical Sciences, Department of Clinical Analysis, Federal University of Paraná, Curitiba 80210-170, PR, Brazil; christiane.lima@ufpr.br (C.M.G.d.L.); nathaliamarcallo@gmail.com (N.M.P.S.); p.rothbarth@hotmail.com (P.R.S.); dayaneaguiar@ufpr.br (D.F.A.); luanamota@ufpr.br (L.M.F.); volanski@gmail.com (W.V.); geraldopicheth@gmail.com (G.P.); 2Pharmacy Course, Federal University of Paraná, Curitiba 80210-170, PR, Brazil; rafaelacirillo@ufpr.br

**Keywords:** hyperglycemia, diabetic retinopathy, angiogenesis, gene polymorphism

## Abstract

**Background/Objectives**: Type 2 diabetes mellitus (T2DM) is a chronic and multifactorial metabolic disorder associated with genetic and environmental factors. Vascular endothelial growth factor (VEGF) plays a crucial role in angiogenesis and vascular homeostasis, and genetic polymorphisms in the *VEGF* signaling pathway have been linked to the T2DM development, progression, and complications. This scoping review investigated the association between *VEGF* gene and *VEGF* receptors single-nucleotide polymorphisms (SNPs) and susceptibility to T2DM and vascular complications. **Methods**: A thorough systematic review was performed utilizing scientific databases (PubMed, Web of Science, and Scopus) in March 2025. From an initial pool of 796 records, 59 relevant articles were selected for inclusion in the analysis. **Results:** The most frequently studied SNPs were rs2010963 (31/59), rs699947 (16/59), rs3025039 (15/59), rs833061 (11/59), rs1570360 (7/59) in the *VEGFA* gene and rs2071559(6/59) in *VEGFR2*. The studies include a diverse range of ethnic groups, including Asian, European and Middle Eastern populations. The main complications associated with these SNPs were microvascular conditions such as diabetic retinopathy (DR) (49/59), diabetic neuropathy (DPN) (6/59), diabetic nephropathy (DNP) (2/59), and as well as macrovascular complications including diabetic foot ulcers (DFU) (10/59). The results revealed that these polymorphisms, particularly rs3025039 and rs2010963, were more consistently associated with microvascular complications such as DR rather than with T2DM itself. The C allele of rs2010963 was associated with increased risk of DR in Indian populations, while no such association was observed in European. Similarly, the T allele of rs3025039 conferred protection against DPN in a Chinese population but was associated with higher DR risk in an Indian study, suggesting that the same allele may play distinct roles depending on ethnic background and clinical phenotype. **Conclusions**: *VEGF* signaling pathway genetic polymorphisms demonstrate potential as biomarkers for diabetic complications, especially microvascular outcomes. The findings suggest a genetic basis for differences in complications of T2DM. Future studies should investigate relevant SNPs across diverse ethnic groups to better understand genetic risks associated with the disease and its vascular complications.

## 1. Introduction

The vascular endothelial growth factor (VEGF) family regulates angiogenesis and lymphangiogenesis [1] while also contributing to processes such as lipid metabolism [2], inflammation [3], and oxidative stress [4]. The VEGF family consists of several secreted dimeric glycoprotein growth factors: VEGFA, VEGFB, VEGFC, VEGFD, VEGFE (viral VEGF, in parapoxvirus 1), VEGF-F (snake venom VEGF), placenta growth factor (PIGF), and EG-VEGF (Endocrine gland-derived vascular endothelial growth factor). VEGF receptors, which bind to tyrosine kinase receptors (VEGFR1 and VEGFR2), are mainly expressed on vascular endothelial cell [5,6,7], while VEGFR-3 is mostly expressed on lymphatic endothelial cell [8]. VEGFR1 (FLT-1) primarily interacts with VEGFA and VEGFB, impacting vascular permeability and endothelial cell survival through the PI3K/Akt/mTOR pathway. VEGFR2 (KDR/Flk-1) is mainly activated by VEGFA, triggering angiogenic signaling via pathways like PLCγ, Ras/Raf/MEK/ERK, and PI3K/Akt that regulate endothelial cell proliferation, survival, migration, and permeability [9,10]. VEGFR3 (Flt-4) has an affinity for VEGFC and VEGFD [9,11,12], influencing lymphatic endothelial cells differentiation, tubulogenesis, proliferation (mitogen effect), migration and survival of lymphatic endothelial cells [9,12]. Accordingly, Figure 1 illustrates the molecular mechanisms triggered by VEGF signaling, emphasizing receptor-ligand interactions, and the downstream signaling cascades that lead to angiogenesis and lymphangiogenesis [4,13].

The actions of VEGF are not limited to the vascular system; VEGF plays a role in bone formation [14], hematopoiesis [15], wound healing [16], development [17], immune homeostasis and lipid transport [18]. Nevertheless, VEGF signaling has also been reported to modulate pathological angiogenesis, such as inflammatory diseases [19], vascular complications associated with diabetes [20,21], diabetes-induced ocular neovascularization [22]. Furthermore, there is evidence that lymphatic vasculature dysfunction is involved in the pathogenesis of obesity and obesity-associated dyslipidemia and low-grade chronic inflammation [23,24]. Several lines of evidence suggest that the VEGF–VEGFR2 signaling axis is at least partially inhibited in diabetes [25,26,27,28]. The VEGFA expression was downregulated in the obese with type 2 diabetes mellitus (T2DM) [29]. Therefore, current evidence collectively indicates a context-dependent dysregulation of the VEGF–VEGFR2 axis in type 2 diabetes. In metabolic and vascular tissues, studies report diminished VEGFA expression and impaired VEGFR2 activation, characterized by reduced phosphorylation, altered receptor internalization, and impaired downstream signaling, which can lessen pathway activity. In contrast, in the retina, VEGF is often upregulated in cases of diabetic retinopathy, contributing to pathological angiogenesis.

Obesity and T2DM often manifest with persistent inflammation and dysfunction in adipose tissue [30]. T2DM is the most prevalent form of diabetes mellitus (DM), a chronic disease characterized by hyperglycemia due to secretion deficiencies and resistance to insulin [31]. In 80–90% of cases, T2DM is associated with excess weight, sedentary lifestyle, poor dietary habits, hypertension, and dyslipidemia. Thus, the disease depends on environmental and genetic factors. Insulin resistance, a hallmark of T2DM, leads to impaired glucose uptake by cells, causing elevated blood sugar levels and long-term complications [32].

Microvascular and macrovascular complications are the leading causes of morbidity and premature mortality among individuals with T2DM [33]. Despite adequate long-term glycemic control, the disease frequently leads to severe clinical outcomes, including diabetic retinopathy (DR), nephropathy (DNP), and neuropathy (DPN), which represent the main microvascular complications. In parallel, macrovascular complications such as cardiovascular disease (CVD), myocardial infarction (MI), stroke, and diabetic foot ulcers (DFU) significantly contribute to disability and reduced life expectancy. These complications, driven by chronic hyperglycemia-induced endothelial dysfunction and inflammation, continue to negatively impact and serious challenges to the clinical management and quality of life of diabetic patients [33]. Growth factors like VEGF are essential in modifying and accelerating tissue damage induced by hyperglycemia, which is a key risk factor in diabetic complications [34], and there have been many reports on the role of VEGF-mediated angiogenesis and vascular permeability changes in chronic microvascular and macrovascular complications of diabetes [35].

The genes encoding VEGF and its associated receptors, which are involved in the angiogenesis pathway, exhibit polymorphism [36,37]. Polymorphisms are recognized for their role in modulating gene expression and contributing to phenotypic variability, which may increase susceptibility to different diseases or modify responses to environmental factors [38]. Some studies have shown that specific polymorphisms in the *VEGF* gene family can modulate individual susceptibility to vascular complications (for review see [39]) and T2DM [40,41]. These genetic variants may impact VEGF expression, alter receptor binding affinities, or modulate downstream signaling pathways. Consequently, this can change vascular permeability, angiogenic potential, and the inflammatory response, all of which are essential mechanisms involved in the pathogenesis of diabetic complications. These processes are intricately linked to the pathophysiology of T2DM and its associated complications [42,43].

Some polymorphisms are associated with increased risk of complications development, contributing to higher incidence or severity of DR, DNP or DPN. Remarkably, others appear to have protective effects, reducing the likelihood or progression of such outcomes [44,45]. The direction and strength of these associations often vary depending on the ethnic background, allelic distribution, and specific complication analyzed, highlighting the complexity of the genotype–phenotype relationship in DM [20]. Identifying these polymorphisms as either genetic risk markers or protective can aid in clarifying the genetic architecture of diabetes and offer promising avenues for early detection, patient stratification, and precision medicine [46,47].

Therefore, this scoping review aims to provide an overview of the association between polymorphisms in the *VEGF* genes and their receptors with T2DM and its microvascular and macrovascular complications. Furthermore, this review seeks to highlight how genotyping studies of these polymorphisms contribute to understanding the genetic predisposition and development of these conditions, offering a detailed analysis of the identified polymorphisms and emphasizing potential targets for future research. This could ultimately facilitate a more personalized approach to the clinical management of these diseases.

## 2. Methods

The current review followed the recommendations outlined by the Joanna Briggs Institute [48] and was reported in accordance with the PRISMA Extension for Scoping Reviews (PRISMA-ScR) guidelines [49]. To ensure thorough documentation, the study’s protocol was registered on the Open Science Framework (OSF) and are available at https://doi.org/10.17605/OSF.IO/MSHVQ. The study followed a structured and transparent process, which included developing a detailed search strategy, applying it across multiple databases, selecting studies, extracting data, and synthesizing results. A PRISMA-ScR checklist is provided as Appendix A.

### 2.1. Research Strategy

The research was conducted across the PubMed, Scopus, and Web of Science databases (Mach 2025), without temporal or linguistic restrictions (Table 1). Each database applied a tailored search strategy, combining descriptors related to T2DM, the *VEGF* gene and its receptors, and associated complications using Boolean operators ‘OR’ and ‘AND.’ A manual search was also conducted by reviewing the reference lists of included studies and performing a simple search on academic sites, ensuring that relevant studies beyond the indexed articles were identified and assessed.

### 2.2. Eligibility Criteria

The eligibility criteria were based on a conceptual framework focused on polymorphisms in the *VEGF* gene and receptors in T2DM and its associated complications. The central aim was to explore the role of genetic polymorphisms in susceptibility to T2DM and its commonly associated complications. We established eligibility criteria using the PCC approach. Population: individuals diagnosed with T2DM. Concept: SNPs in genes related to the VEGF signaling axis, including VEGFA, VEGFB, VEGFC, VEGFD, and their receptors VEGFR1/FLT1, VEGFR2/KDR, and VEGFR3/FLT4. These SNPs were evaluated for their associations with T2DM and/or its complications. Context: human genetic association studies (case–control, cross-sectional, or cohort) that report genotypes and genotype–phenotype relationships in clinical or population settings. Therefore, the guiding research question was: “***Are polymorphisms in the VEGF gene and its receptors associated with susceptibility to T2DM and its complications*?**”. Only studies that focused on the genotyping of polymorphisms in the *VEGF* gene and its relation to T2DM or its complications were included. Articles based on animal studies, those lacking genotyping frequency data, or those not related to T2DM and its complications were excluded. Additionally, studies examining SNPs in genes other than *VEGF* (*A*, *B*, *C*, *D*), *VEGFR1*, *VEGFR2*, or *VEGFR3* were also excluded. Review articles, editorials, books, conference abstracts, and articles written in non-Roman characters were not considered.

### 2.3. Study Selection

The article selection process began with a search across the previously mentioned databases. All search results were imported into the Rayyan web platform, where duplicates were eliminated. Two reviewers independently screened the titles and abstracts through a blind evaluation. Articles that met the inclusion criteria were thoroughly reviewed in full to assess their suitability based on predetermined eligibility criteria. Any disagreements were resolved with the help of a third reviewer. Articles that did not meet the eligibility criteria were excluded from further consideration, and the reasons for their exclusion were meticulously documented (Appendix A).

The identification, selection, eligibility criteria, and the outcome (number of studies included and excluded) of the article election process were reported in a flowchart provided by the PRISMA-ScR platform (Figure 2).

### 2.4. Data Extraction

Relevant data from the selected studies were compiled and presented in a detailed table (Table 2), encompassing details such as the first author, publication year, country of origin (corresponding authors), participant ethnicity, gene, the total number of participants (both cases and controls), the polymorphisms examined, methodologies utilized, and the related complications. To improve clarity and facilitate interpretation, key interrelationships identified across studies were visually depicted in graphical formats, based on a descriptive statistic (Figure 3).

A narrative synthesis was conducted to elucidate the key findings, facilitate comparative analyses across studies, and examine possible explanations for identified discrepancies or gaps. Subsequently, the data underwent a systematic review, incorporating collaborative discussions to critically evaluate the study outcomes.

## 3. Results and Discussion

### 3.1. Characteristics of Included Studies

The search yielded 796 references that after removing duplicates and applying inclusion and exclusion criteria, 517 articles were selected (Figure 2). An initial screening of titles and abstracts resulted in the exclusion of 413 papers, leaving 104 articles for full-text review. Of these 104 articles, we excluded 54 given specific reasons that are presented in the flowchart and detailed described in Appendix A. Finally, a manual search yielded 9 articles, and 8 were excluded. Applying this study’s inclusion criteria, this review included a total of 59 studies.

The selected publications encompass a temporal range between 2006 and 2024, with a noticeable concentration of studies in recent years; 13 studies were included that were published between 2021 and 2023 (Figure 3A). A total of 20 SNPs were identified across the included studies. The most frequently studied SNPs were rs2010963 (n = 31/59), 27), rs699947 (n = 16/59), rs3025039 (n = 15/59), rs833061 (n = 11/59), rs1570360 (n = 7/59), rs2071559 (n = 6/59), rs13207351 (n = 4/59), rs2305948 (n = 3/59), rs25648 (n = 2/59), rs6921438 (n = 2/59), rs2146323 (n = 2/59), rs10434 (n = 2/59) and rs3025021 (n = 2/59). Several SNPs were assessed in only one study, including rs10738760, rs3025035, rs833069, rs12366035, rs7664413, rs7993418 e rs3025020. In total, 13 SNPs are related to *VEGF* genes (rs10738760, rs12366035, rs6921438, rs3025020, rs10434, rs25648, rs2146323, rs13207351, rs1570360, rs833061, rs3025039, rs699947, rs2010963), and 3 variants map to *VEGFR* genes—rs7993418 in *VEGFR1*, and rs2305948 and rs2071559 in *VEGFR2*. (Figure 3B). Among these, 3 SNPs (rs10738760, rs3025035 and rs833069) did not show any statistically significant association with T2DM or its complications.

The complications associated with these SNPs were DR (49/59), T2DM (18/59), DFU (10/59), DPN (6/59), DNP (2/59), MI (2/59), DN (2/59), CRI (1/59) and CAD (1/59) (Figure 3C). This distribution demonstrates the predominant focus on microvascular complications. The distribution of SNPs by gene, with the predominance of *VEGFA* (57/59) and *VEGFR2* (7/59) in the studies reviewed is largely explained by their central role in the VEGF signaling pathway, which is critically involved in the vascular changes associated with diabetes and its complications (Figure 3D). VEGFA encodes the most potent angiogenic factor in this family and is strongly upregulated in response to hyperglycemia, hypoxia, and inflammation, conditions commonly present in diabetic tissues [103]. Its expression promotes endothelial cell proliferation, migration, and increased vascular permeability, key events in the development of DR, DNP, and DPN. VEGFR2 is the main receptor mediating VEGFA activity, which regulates angiogenesis, inflammation, and endothelial function. Due to their pivotal position in this axis, polymorphisms in *VEGFA* and *VEGFR2* genes are highly relevant, as they can directly affect VEGF expression or receptor binding affinity, thereby altering the biological response and contributing to individual susceptibility to diabetic complications [104].

Figure 3F illustrates the association between individual SNPs and diabetic complications. Red dots represent SNPs associated with increased risk, green dots indicate protective associations, and yellow dots denote SNPs with mixed associations depending on the complication. The absence of markers for some SNP-complication pairs indicates lack of statistically significant association. This pattern reflects a growing body of research as our understanding of the genetic predisposition to T2DM continues to evolve. In terms of the populations studied, the articles included in this review represent a diverse array of ethnic groups and geographical backgrounds (Figure 3E). The research spans various countries, including China (13/59), Egypt (7/59), India (6/59), Iran (5/59), Slovenia (4/59), Pakistan (3/27), Poland (2/59), Korea (2/59), Indonesia (2/59), Mexico (2/59) and others, highlighting the global nature of studies on *VEGF* polymorphisms in relation to T2DM. This diversity among study populations facilitates a more comprehensive understanding of how genetic variations may impact the development of T2DM and its associated complications across different ethnicities, providing valuable insights into potential genetic risks and regional differences in disease manifestation.

The studies included in this scoping review employed various genotyping techniques to analyze *VEGF* and receptor polymorphisms (Figure 3G). The most frequently used method was PCR-RFLP, which was applied in 34 studies and is a widely used and cost-effective technique for SNP detection. TaqMan assays (11/59) provide high specificity and efficiency for large-scale genotyping [105]. ARMS-PCR (5/59) allows allele-specific amplification without restriction enzymes [106]. The iPLEX (3/59) is a mass spectrometry-based method that enables multiplex SNP genotyping with high precision and low DNA input [107]. PCR/LDR (2/59) enhances SNP detection through a ligase-based reaction. Additionally, KASP (2/59) is a fluorescence-based competitive PCR technique known for its accuracy in genotyping [108]. Lastly, one study applied RT-PCR methodology, a highly sensitive quantitative method for SNP and gene expression analysis [109] and 1 study report using PCR follow by direct sequencing (PCR/SEQUENCED). These methodological variations highlight the importance of standardization for future research, ensuring comparability and reproducibility across different studies and populations.

Excepting rs1570360 and rs2010963, both in the *VEGFA* gene, no SNPs showed deviation from Hardy–Weinberg equilibrium (HWE). The fact that most SNPs were in HWE support the reliability of the findings. However, the deviation observed in rs1570360 and rs2010963 may indicate potential genotyping errors, population structure issues, or sample selection bias. Another possibility is that this deviation happens for biological reasons, such as natural selection, interactions between genes, linkage disequilibrium or a real association with the studied condition. Therefore, these findings should be interpreted carefully to distinguish between technical issues and true genetic effects [110].

### 3.2. Association Between Polymorphisms in the VEGF Gene and VEGFR2 and the Occurrence of TDM2 or Related Complications

To comprehend the relationship between *VEGF* gene polymorphisms and T2DM, it is essential to examine the most studied genetic variants. In this review, we identified in scientific literature the most frequent SNPs related to *VEGF* gene and T2DM aiming to facilitate a thorough evaluation of their potential role in disease susceptibility and its associated complications.

Among the most frequently studied SNPs, those in *VEGFA* (e.g., rs3025039, rs2010963, rs699947, rs1570360, rs833061, rs13207351) and *VEGFR2* (e.g., rs2071559 and rs2305948) stand out (Table 3). These polymorphisms have been examined in multiple populations and have been associated with variations in VEGF expression, vascular dysfunction, and susceptibility to microvascular and macrovascular complications of diabetes.

To provide a clear visualization of the studied genetic variants, Figure 4 illustrates the approximate locations of the selected SNPs within the *VEGFA* and *VEGFR2* genes. This figure highlights key regions such as the promoter, 5′ untranslated region (UTR), 3′ UTR, exonic and intronic regions, which may influence gene expression and protein function. SNPs identified in only one study (rs10738760, rs3025035, rs833069, rs12366035, rs7664413, rs7993418 and rs3025020) are not shown in the figure but are addressed in detail in the discussion section, as their limited occurrence reduces the strength of association and generalizability.

By structuring the analysis based on study prevalence and visualizing SNPs distribution within the genes, this review aims to highlight the most extensively explored genetic variants and provide insights into their potential role in diabetes pathophysiology. The upcoming sections provide an overview the characteristics of the included studies and discuss the associations identified between these polymorphisms and diabetes-related complications.

#### 3.2.1. *VEGFA* Polymorphisms

The human *VEGFA* gene is located on chromosome 6 in region 6p21.3 and organized into eight exons separated by seven introns [111]. It gives rise to multiple VEGF isoforms via alternative exon splicing [112,113] and various post-transcriptional mechanisms [114,115,116]. VEGFA exists in multiple isoforms of variable exon content and exhibits strikingly contrasting properties and expression patterns [112]. The gene is highly polymorphic, with 30 functional SNPs in the promoter region, 5′ UTR, and 3′ UTR linked with altered VEGF protein expression [27,117,118,119,120,121,122,123,124], and VEGF protein levels have been shown to be elevated in PDR patients and the vitreous of subjects with diabetes [125,126,127], T2DM patients without and with mixed microvascular complications [95,128,129], diabetic polyneuropathy [130], diabetic retinopathy [44,70], diabetes and hyperglycemia [131], overweight and obese individuals [132], metabolic syndrome [133].

##### *VEGFA* rs2010963

The rs2010963 (−634G/C) polymorphism, located in the 5′ UTR of the *VEGFA* gene, involves a G/C substitution at position −634 and has been reported to influence VEGF expression levels. In vitro studies showed that the C allele increases VEGFA transcription and translation [119,120,134], while individuals homozygous for the G allele tend to have lower serum VEGF concentration [27,118,135,136]. However, VEGFA is expressed in numerous cell types and there are several other polymorphisms that exist in the 5′-flanking regions of the *VEGF* gene [120], therefore it will require further study to clarify the influence of the polymorphisms on VEGF expression.

This variant has been widely investigated, with heterogeneous results depending on the population and phenotype assessed. In this review, rs2010963 was the most frequently investigated SNP (31/59 studies), predominantly in association with DR, including both NPDR and PDR forms. DR is a progressive microvascular complication of DM and an essential cause of acquired blindness. There is growing evidence implicating genetic factors in susceptibility to diabetic retinopathy. Angiogenic factors, such as VEGF, are involved in the pathogenesis of DR [137]. It is classified into two main stages: NPDR and PDR. The first stage is characterized by microaneurysms, intraretinal hemorrhages, hard exudates, and retinal edema resulting from increased vascular permeability and capillary occlusion. As the disease progresses, ischemia-induced upregulation of VEGF promotes pathological neovascularization, leading to PDR. This advanced stage is marked by fragile new blood vessels that can cause vitreous hemorrhage and tractional retinal detachment, leading to severe vision loss [138,139]. The pathogenesis of DR involves chronic hyperglycemia-induced oxidative stress, inflammation, and neurodegeneration, making tight glycemic control and timely interventions, such as anti-VEGF therapy and laser photocoagulation, essential in preventing vision loss [140].

Multiple studies identified a significant association between rs2010963 and DR, with most implicating the C allele as a risk factor. In the Indian population, Suganthalakshmi et al. (2006) found the GC genotype increased DR risk (OR = 2.33; 95% CI: 1.24–4.36) [50], while Uthra et al. (2008) reported an even higher risk in patients with microalbuminuria (OR = 8.9; 95% CI: 1.4–58.3) [55]. Choudhury et al. (2015) also associated the C allele with PDR occurrence [44]. In Egyptian populations, El-Shazly et al. (2014) observed that the CC genotype was significantly associated with diabetic macular edema (*p* < 0.001), and Kamal et al. (2016) found the C allele to be more frequent among PDR patients compared to NPDR and controls, suggesting a role in disease progression [69,79].

In East Asian populations, Yang et al. (2010) reported an association between the CC genotype and DR in Chinese patients, while Chen et al. (2016) confirmed that the C allele correlated with DR and higher serum VEGFA levels [61,77]. Conversely, Jin et al. (2021) found the CG genotype to be protective against PDR in Chinese patients (OR = 0.588; 95% CI: 0.366–0.946), suggesting potential population-specific effects [91].

In European ancestry populations, results have been mixed. Errera et al. (2007) identified the CC genotype as an independent risk factor for PDR in Brazilian patients of European ancestry (OR = 1.9; 95% CI: 1.01–3.79; *p* = 0.04) [51], while Szaflik et al. (2007) observed that the C allele increased DR risk and VEGFA promoter activity in Polish patients [53]. However, studies by Buraczynska et al. (2007) in Poland and Petrovic et al. (2008) in Slovenia found no association with DR or PDR [34,54]. Similarly, studies in Korea [60], South Africa [72], Mexico [141], and other Chinese studies [61,70] reported no significant association with DR, highlighting regional differences in genetic background and environmental exposure on the effects of rs2010963. Therefore, similar associations with the C allele or CC genotype were reported in Indian, Egyptian, and Chinese cohorts, whereas other studies in European, African, and Asian populations found no association or even protective effects. This highlights the population-specific and heterogeneous nature of rs2010963 associations.

Although rs2010963 emerged as the most frequently investigated SNP in this review, particularly in relation to DR, it was not the exclusive focus of all studies. This polymorphism has also been examined for associations with other diabetic complications, including MI, DFU, DNP, DPN.

For MI, the major macrovascular complication of diabetes characterized by plaque rupture, thrombosis, and myocardial necrosis, with VEGFA implicated in vascular remodeling, endothelial dysfunction, and plaque instability contributing to its pathogenesis [142], Petrovic and collaborators (2007) reported that the CC genotype increased MI risk in a study conducted in a Slovenian population with T2DM (OR = 2.1; 95% CI = 1.1–3.9; *p* = 0.019) and might be used as a genetic marker of MI, alongside higher serum VEGFA levels in CC [52].

Li and coworkers (2018) found that the CC genotype was less frequent among DFU patients (OR = 0.36; 95% CI: 0.17–0.77; *p* = 0.008), suggesting a protective effect likely mediated by enhanced VEGFA-induced angiogenesis promoting wound healing [85]. DFU is a debilitating and complex complication characterized by chronic, non-healing foot ulcers resulting from neuropathy and peripheral vascular disease, where angiogenesis plays a crucial role in tissue repair and that occurs predominantly in individuals with long-standing or poorly controlled DM. Prolonged hyperglycemia leads to metabolic and vascular alterations, including endothelial dysfunction, oxidative stress, and impaired immune response, which together compromise tissue perfusion and wound healing [85].

DNP is a complication of DM that affects the renal microvasculature. In individuals with diabetes, there is an increase in the glomerular filtration rate, or hyperfiltration, which is driven by a greater relaxation of the afferent arterioles compared to the efferent arterioles. This leads to elevated blood flow through the glomerular capillaries, increasing pressure. Over time, these conditions result in glomerular hypertrophy and an expansion of the glomerular capillary surface area. These hemodynamic alterations contribute to the disease’s onset and progression [143,144]. Nikzamir and collaborators (2012) reported that the GG genotype increased DNP risk (OR = 1.77; 95% CI: 1.12–2.79; *p* = 0.014) in an Iranian cohort, whereas Buraczynska and colleagues (2007) found no significant association in patients from Poland [34,66].

Luo and collaborators (2019) reported a slightly increased risk with the G allele (OR = 1.15; 95% CI: 1.03–1.30) in Chinese patients, although overall evidence for rs2010963 in DPN remains limited [36]. DN is a late and prevalent complication of diabetes and can often be seen in T2DM at the time of diagnosis. DN is among the most common long-term complications of diabetes, which affects up to 50% of patients [92]. It is defined as signs and symptoms of peripheral nerve dysfunction in diabetics. It is progressive and irreversible and is one of the leading causes of ulceration and amputation of the lower limbs [75]. It encompasses a group of changes related to the structural and functional involvement of motor, sensory, and autonomic fibers. The pathogenesis remains under investigation; however, current hypotheses point to neural alterations from hyperglycemia and endothelial dysfunction as key contributors. Results from absolute or relative ischemia of endoneural and epineural vessels, leading to thickening of the basement membrane, decreased blood flow, and changes in vascular permeability. Due to this series of factors, microvascular perfusion, insufficient flow, and neural injury occur [145,146].

In summary, rs2010963 appears to contribute to the risk of PDR and DR in specific populations through increased VEGFA expression mediated by the C allele. Interestingly, the same variant that promotes pathological angiogenesis in DR may also exacerbate vascular complications in coronary arteries, increasing susceptibility to MI. Conversely, it may exert a protective effect in DFU by enhancing wound healing via angiogenesis, while its associations with T2DM itself, DNP, or DPN remain inconclusive. The variability in findings across studies highlights the complex interplay between genetic background, environmental factors, and disease phenotype, underscoring the need for further large-scale, multi-ethnic studies and functional analyses to clarify its mechanistic pathways and potential clinical significance in risk stratification and personalized management of diabetic complications.

##### *VEGFA* rs699947

The rs699947 (−2578C/A) polymorphism, located at the −2578 translation start sites in the promoter region of the *VEGFA* gene [147], and is thought to be responsible for differentiated gene expressions. The −2578 A allele was associated with decreased [117,118] and increased [122] VEGF synthesis, suggesting complex regulatory effects. The rs699947 polymorphism has been investigated for its role in diabetes and related complications, with studies revealing diverse associations and potential functional implications, in this review 16 of 59 studies are related with this SNP.

For DFU, several studies have highlighted a protective effect of the A allele. Amoli et and collaborators (2011) observed that the AA genotype was less frequent among DFU patients in Iran [63]. Similarly, other study reported that the A allele was significantly associated with a reduced risk of diabetic foot complications in Italian patients, while the CC genotype was linked to increased VEGFA expression, potentially exacerbating vascular damage [98]. The protective association of the A allele is further substantiated by earlier research, which shows that Chinese individuals carrying this allele exhibit a decreased susceptibility to DFU (OR = 0.55; 95% CI: 0.34–0.89) [148].

Regarding DR, findings are more heterogeneous. Chun and colleagues (2010) in Korea reported that the A allele increased DR risk, with the C allele possibly exerting a protective role [60]. Qayyum and coworkers (2023) in Pakistan also linked the A allele to elevated DR risk [100]. Conversely, other studies found no significant association in Egypt, Poland, and China populations, highlighting potential ethnic, environmental, or methodological influences on these divergent results [34,61,78].

Beyond microvascular complications, in the context of T2DM and its systemic manifestations, Elfaki and collaborators (2021) in Sudan found that individuals with the CA genotype had altered lipid profiles and increased cardiovascular risk, suggesting a broader metabolic impact of this SNP beyond local vascular complications [40].

In summary, the rs699947 variant demonstrates population and phenotype specific associations with diabetic complications. The A allele is consistently reported as protective against DFU in multiple studies, whereas its role in DR remains controversial, with studies showing both risk and protective effects. The CC genotype, associated with higher VEGFA expression, may predispose individuals to vascular damage, whereas the CA genotype has been implicated in dyslipidemia and cardiovascular risk. These findings highlight the complex interplay between genetic variants and diabetes-related complications, reinforcing the need for further functional studies to elucidate the mechanisms underlying these associations.

##### *VEGFA* rs3025039

The rs3025039 (+936 C/T) is in the *VEGFA* 3′ untranslated region (3′ UTR) and involves a variation in C/T at position +936, which may alter the expression of the VEGFA as it is a critical regulatory region influencing mechanisms of mRNA generation [149,150]. The C allele has been associated with increased mRNA expression and, consequently, higher plasma VEGFA levels. In contrast, the T allele is linked to reduced VEGFA concentrations as a result of the loss of a potential binding site for the transcription factor AP-4 [56,124,151].

The rs3025039 polymorphism has been widely investigated in relation to diabetes and its complications, with varying results across different populations among 15 of 59 studies concentrating on T2DM and associated complications, but predominantly assessing its association with DR.

The T allele demonstrated heterogeneous associations. In DR, a study reported the T allele and TT genotype as risk factors associated with higher VEGF levels, while Choudhury and colleagues (2015) found the T allele increased PDR risk in Indian patients [44,57]. Similarly, another report observed the T allele as a risk factor for NPDR [41]. Conversely, other studies reported no significant association with DR, T2DM, PDR or neuropathy, highlighting potential ethnic or sample size influences [55,70,75,94]. For DPN, studies consistently reported a protective effect of the T allele or CT+TT [45,83,86]. In DFU, no significant association was reported [98].

Regarding T2DM, Imbaby and colleagues (2021) found the CT genotype associated with increased T2DM risk in Egyptian patients, although no association was observed with DPN [92]. Additionally, other study reported that both the CT genotype and T allele had higher frequencies in groups with mixed diabetic complications, suggesting potential broader vascular effects [95].

Overall, the T allele tends to be protective in DPN and some DR studies, but appears to increase risk in PDR, NPDR, and certain DR populations, highlighting population-specific and phenotype-dependent effects. Its functional link to reduced VEGF expression suggests it may modulate angiogenic responses differently across tissues and disease stages.

In summary, while the rs3025039 variant demonstrates inconsistent associations with T2DM and its complications across studies, it appears to play a more pronounced role in microvascular complications such as PDR and DR. The T allele and TT genotype may confer risk or protection depending on the specific complication and population, underscoring the need for further research to clarify these relationships and their underlying mechanisms.

##### *VEGFA* rs833061

The rs833061 (−460C>T) polymorphism is a C>T SNP at position −460 of the *VEGFA* gene promoter and it has been related to higher VEGF expression [152,153]. In this review, 11 studies investigated rs833061, predominantly in relation to DR, but also in DFU and chronic renal insufficiency (CRI).

Khan and colleagues (2020) demonstrated a significant association between the rs833061 variant and the early stages of DR in Pakistani patients, identifying the C allele as a major risk factor. This allele is associated with increased VEGF activity, which may drive the development of microvascular complications by enhancing angiogenesis and vascular permeability [89]. Similarly, other study confirmed a link between rs833061 and DR, though no direct association with T2DM was observed [99]. Their findings showed that the C allele upregulates promoter activity, while the T allele is linked to an overexpression of the *VEGFA* gene. These differential effects highlight the complex regulation of VEGF expression and suggest that both alleles influence VEGF levels, although mediated by distinct mechanisms. Such variations in VEGF expression are likely contributors to the pathogenesis of DR. In China, a significant association of the CC genotype with DR (OR = 3.72) was reported by Yang and collaborators (2011) [61], which was further corroborated by the study of Yang and colleagues (2014) (*p* = 0.001) [73]. In a population from India, Paine and collaborators (2012) also observed an increased risk of PDR in the CC genotype (OR = 3.66), suggesting this variant may influence progression to more severe retinopathy forms [67,73]. In contrast, other studies performed in populations from Poland and China did not observe significant associations with DR, highlighting possible differences due to ethnicity, environment, or study design [53,70]. Interestingly, Yuan and colleagues (2014) found the C allele to be protective against NPDR, suggesting its effects may vary across retinopathy stages [53,70,74].

For DFU, results were mixed. Zhuang et al. (2017) (China) reported a significant association between rs833061 and DFU risk [82]. Dahlan et al. (2019) (Indonesia), however, found no association, indicating possible population-specific effects [87].

Regarding chronic renal insufficiency (CRI), Tiwari et al. (2009) (India) found the CT genotype significantly increased CRI risk (OR = 2.23), suggesting a broader role for rs833061 beyond ocular complications [59,87].

Overall, the C allele appears to increase VEGF expression, enhancing angiogenesis and vascular permeability—key mechanisms driving DR and potentially other vascular complications in diabetes. Some studies also suggest the T allele may promote VEGFA overexpression through distinct pathways, highlighting complex regulatory dynamics where both alleles could modulate disease risk differently depending on context.

In summary, the rs833061 polymorphism in the *VEGFA* promoter region is closely associated with the early stages of DR, with the C allele acting as a risk factor by upregulating VEGF activity. The T allele, through overexpression of VEGFA, may also play a significant role in the progression of DR. These findings underscore the critical role of VEGFA regulation in the development of DR and suggest that rs833061 could serve as a valuable biomarker for early detection and risk stratification in DR. However, further research is crucial to fully understand the molecular mechanisms through which this variant modulates VEGF expression and its broader implications for diabetic complications.

##### *VEGFA* rs1570360

The rs1570360 (−1154G/A) polymorphism, located in the promoter region of the *VEGFA* gene, involves a G/A substitution at position −1154 [147]. The A allele was associated with low VEGF expression [117,118,119,122], and the AA genotype was associated with poor vascularization [123]. One of the revised studies proposed through in silico analysis suggested that the A allele could potentially affect VEGF expression by creating a binding site for the GABP transcription factor, which could lead to lowered VEGF expression [154]. This SNP has been investigated in 7 studies included in this review, predominantly assessing its association with DR and its proliferative form (PDR).

Among these studies, Choudhury and collaborators (2015), conducted in an Indian population, reported a significant association between the AA genotype and increased risk of PDR, suggesting that this variant could enhance VEGFA expression, and promote pathological angiogenesis characteristic of advanced DR [44]. Conversely, Fan and colleagues (2014) found no association between rs1570360 and DR in Chinese patients, although they noted elevated VEGFA serum levels in DR patients, suggesting a possible regulatory influence of promoter variants [70].

Similarly, Chun et al. (2010) in Korea, Chen et al. (2019) in Taiwan, Khan et al. (2020) and Qayyum et al. (2023) in Pakistan reported no significant association between rs1570360 and DR or T2DM, highlighting inter-population differences [60,89,100,155]. Additionally, other study evaluated rs1570360 in relation to DFU in a Chinese cohort, finding no significant association [82].

These findings suggest that while the AA genotype may be a risk factor for PDR in specific populations, particularly in India, its role remains inconclusive across other ethnic groups, and it shows no significant association with DFU. This variability may reflect differences in genetic background, linkage disequilibrium with other functional variants, environmental influences, or limited statistical power in individual studies. Additionally, other contributing factors include small sample sizes (leading to insufficient statistical power), variability in complication type, and the lack of correction for multiple testing when assessing associations with more than one SNP. These issues increase the risk of chance correlations, which must be interpreted considering biological plausibility and the functional relevance of the variant studied [156]. Variants in the promoter region, such as rs1570360, can influence transcription factor binding, leading to altered VEGFA expression levels. Increased VEGFA expression promotes angiogenesis and vascular permeability, key processes in developing PDR and other microvascular complications [117].

##### *VEGFA* rs13207351

The rs13207351 (−1190A/G) polymorphism is located at the promoter region of the *VEGFA* gene and involves an A/G substitution [157], which may hold the capacity to regulate gene expression level. The rs13207351 was found to create conserved punitive transcriptional factor binding sites (TFBS). The G allele from the rs13207351 polymorphism is predicted to create a unique transcription factor binding site (TFBS) for NFIC, whereas the A allele generates two distinct TFBS for HINFP and PAX5 [157,158].

This variant has been investigated in 4 studies included in this review. In a study conducted in a Chinese population, the AA genotype was significantly associated with increased DR risk (OR = 3.76, 95% CI: 1.21–11.71), suggesting that this promoter variant may enhance VEGFR2 expression, thereby potentiating VEGF signaling and pathological angiogenesis characteristic of DR [64]. Similarly, other investigation confirmed a significant association with DR (*p* < 0.001) in Chinese patients, further supporting its potential role as a genetic risk factor [73].

Conversely, Khan and colleagues (2020), found no association between rs13207351 and NPDR, PDR, or T2DM in a Pakistani population, highlighting possible ethnic differences in allele frequencies, linkage disequilibrium patterns, or environmental interactions [89]. Additionally, Li and colleagues (2018) evaluated rs13207351 in relation to DFU in Chinese patients but reported no significant association, suggesting its influence may be limited to retinal vascular pathology rather than peripheral diabetic complications [85].

In summary, the rs13207351 polymorphism of the *VEGFA* gene shows population-specific associations with DR, with the AA genotype conferring increased risk in Chinese cohorts, while no associations have been observed with NPDR, T2DM, or DFU in other populations. Further research, including functional studies and investigations in diverse populations, is needed to fully understand the role of this variant and its potential implications in diabetic complications.

##### *VEGFA* rs2146323

The rs2146323 (+5092C/A) polymorphism is an intronic variant located 111 bp 5′ of exon 3 within the *VEGFA* [159]. Although intronic SNPs do not alter protein-coding sequences, they may affect gene expression by influencing mRNA splicing, transcription factor binding, or chromatin structure, thereby potentially contributing to disease pathogenesis [159].

However, functional predictions for rs2146323 suggest a limited regulatory impact. The variant is relatively distant from known transcription factor binding sites, and it does not alter predicted exonic splicing enhancer (ESE) motifs or affect exon splicing. For these reasons, it has been considered a less likely candidate for functional significance. Supporting this, a Finnish case–control study found no association between rs2146323 and the severity of DR [159]

Despite these considerations, two studies included in this review—both conducted in Chinese populations—reported a significant association between rs2146323 and DR. Yang and colleagues (2011) reported a significant association between the AA genotype and increased risk of DR. Their findings suggested that carriers of the A allele, particularly those with the AA genotype, may be more susceptible to the vascular changes underlying DR. This finding was further supported in which was found a higher frequency of the AA genotype among DR patients compared to controls, reinforcing the role of rs2146323 in DR susceptibility [64,73].

Although these findings are limited to a single ethnic group, they suggest that rs2146323 may contribute to the development of diabetic microvascular complications through subtle regulatory effects on VEGFA expression. Additional research in other populations, as well as functional assays, will be essential to clarify the biological significance of this variant.

##### *VEGFA* rs25648

The rs25648 (−7C>T) is in the 5′ UTR of the *VEGFA* gene, positioned near regulatory elements of the gene. According to the Ensemble Genome Browser, this SNP lies within a regulatory region upstream of the transcription start site of *VEGFA*, suggesting a potential influence on gene expression through effects on mRNA stability or translation efficiency [160]. Variations in this region may affect mRNA stability or translational efficiency, potentially influencing angiogenesis, a key pathophysiological mechanism underlying diabetic microvascular complications [161]. Suganthalakshmi and colleagues (2006) investigated this SNP in an Indian population and reported a significant association between the heterozygous genotype and DR, with increased risk for disease development (OR = 4.17; 95% CI: 1.90–9.18; *p* = 0.0001) [50]. These findings suggest a possible involvement of rs25648 in DR susceptibility, potentially mediated by VEGFA-driven angiogenic mechanisms.

Conversely, Amoli and coworkers (2011) evaluated Iranian patients with diabetic foot ulcer (DFU) and found no significant differences in allele or genotype frequencies between cases and controls [63]. This suggests that rs25648 may not play a major role in DFU pathogenesis in this population and highlights the possibility of phenotype-specific effects or ethnic/environmental modulation of its impact.

In summary, while preliminary evidence suggests a possible association between rs25648 and DR, the current data is limited and inconsistent. The lack of association with DFU in another cohort supports the notion of phenotype specificity. Further studies in diverse populations, including functional analyses, are needed to clarify the regulatory role of this SNP in VEGFA expression and its clinical relevance in diabetic complications.

##### *VEGFA* rs10434

The rs10434 (+1612G/A), located in the 3′ UTR in the *VEGFA* gene, may influence mRNA stability and translation efficiency [122], modulate circulating VEGF levels and affect angiogenic processes critical for microvascular complications such as diabetic retinopathy [56].

The G allele of rs10434 was significantly associated with sight-threatening diabetic retinopathy. Specifically, the presence of the G allele conferred a higher risk (OR = 2.6; 95% CI: 1.3–5.3; *p* = 0.002), suggesting that this variant may contribute to increased VEGFA activity and disease severity [56].

Conversely, Fattah and colleagues (2016) found no significant association between rs10434 and DR in an Egyptian population, indicating that the effect of this SNP might be population-specific or modulated by other genetic and environmental factors [78].

In summary, while rs10434 has been implicated in the development of severe DR, the overall contribution remains inconclusive due to conflicting evidence. Further studies in diverse populations are necessary to clarify its role and potential as a genetic biomarker for DR susceptibility.

##### *VEGFA* rs3025021

The rs3025021 polymorphism is in intron 6 of the *VEGFA* gene (NM_003376.5), on chromosome 6p21.1. This variant corresponds to a C/T substitution at genomic position chr6:43750328 (GRCh38/hg38). Although intronic, such variants may influence alternative splicing or interact with intronic regulatory elements that modulate VEGFA expression. These biological processes are central to the pathogenesis of diabetic complications, particularly DR.

This SNP was evaluated in two independent studies, with conflicting results. In a study conducted in an Australian population, the C allele of rs3025021 was significantly associated with blinding DR. The authors reported that individuals carrying the C allele had an increased risk (OR = 3.8; 95% CI: 1.5–10.0; *p* = 0.002), suggesting a possible functional role in modulating VEGF expression and promoting pathological neovascularization in DR [56].

In contrast, in a Mexican cohort, no association between rs3025021 and PDR was observed. The lack of association in this population may be due to genetic heterogeneity, differences in environmental exposure, study design, or sample size limitations [141].

Taken together, these findings indicate that rs3025021 may be a risk marker for severe forms of DR, but its role appears to be population dependent. Further studies are needed to clarify its biological relevance and to assess whether this polymorphism exerts functional effects on *VEGFA* gene regulation or expression.

##### *VEGFA* rs3025020

The rs3025020 (−583C/T) is located in intron 6 region, specifically, is a C/T substitution at position -583 relative to the transcription start site of the *VEGFA* gene [162], and the T/T genotype was associated with increased VEGF serum levels [124,163].

Only one study has evaluated the role of rs3025020 in the context of T2DM and its complications. According to Omar and colleagues (2022), the T allele was associated with increased risk for diabetic complications, with an odds ratio (OR) of 2.67 (95% CI: 1.03–6.91; *p* = 0.04), suggesting that this SNP may be involved in pathways related to angiogenesis and microvascular injury [95].

Although evidence is limited, these preliminary findings indicate that rs3025020 may contribute to the genetic susceptibility to diabetes-related vascular complications. Further studies are warranted to confirm its association and to elucidate its potential regulatory function in VEGFA expression and diabetic pathogenesis.

##### *VEGFA* rs3025035

The rs3025035 polymorphism is in intron 7 of the *VEGFA* gene and involves a C/T substitution. In the only study identified in this review, González-Salinas and collaborators (2016) investigated rs3025035 in a Mexican population and found no significant association between this polymorphism and DR. The lack of association suggests that rs3025035 may not play a major role in the susceptibility to DR in that population [81].

Given the limited evidence and absence of functional studies, the relevance of rs3025035 in diabetic complications remains unclear. Further research, including replication studies in other ethnic groups and functional analyses, is needed to determine whether this variant contributes to VEGFA regulation or microvascular pathogenesis in diabetes.

##### *VEGFA* rs6921438

The rs6921438 located at 171 kb downstream of the *VEGFA* gene, involves a G/A substitution and was associated with circulating VEGF levels and explained 41.2% of the heritability of circulating VEGF levels [129].

According to Bonnefond and colleagues (2013), the G allele of rs6921438, previously linked to higher VEGF levels in the general population, conferred an increased risk of T2DM and DR in the French population. However, this association was not replicated in the Danish population [68]. Similarly, the rs6921438 was not associated with DR in Slovenian patients with T2DM in other two studies [22,164], suggesting potential population-specific genetic effects or the influence of environmental and lifestyle factors on gene–disease associations.

A recent study conducted in a Slovenian population found that the G allele was protective against DN (OR = 0.51, *p* = 0.004), indicating a potentially beneficial vascular role in specific contexts [102]. These conflicting results underscore the complex and possibly context-dependent role of *VEGFA* polymorphisms in diabetic complications.

Taken together, these findings suggest that rs6921438 may influence the risk of diabetic microvascular outcomes in a population-specific manner. The G allele appears to act as a risk allele for T2DM and DR in certain groups, while it may confer protection against DN in others. Further research, particularly functional studies, is warranted to clarify its molecular mechanisms and to determine whether it may serve as a biomarker for disease risk or therapeutic response.

##### *VEGFA* rs833069

The rs833069 (+450T/C) is located in intron 2 of the *VEGFA* gene [165], although intronic, such variants may influence gene expression through mechanisms such as alternative splicing or altered transcriptional regulation, especially in genes with key regulatory functions like *VEGFA*, a major driver of angiogenesis via VEGFR1 and VEGFR2 activation on endothelial cells [166].

Abdelghany and colleagues (2021) demonstrated no significant association between rs833069 and the risk of T2DM or DR in their study population suggesting a limited role in disease progression [90]. Other studies have linked rs833069 with increased risk of DR in patients with type 1 diabetes and with the severity of DR in T2DM [167,168]. These conflicting findings may stem from differences in study design, sample size, ethnic background, or genotyping techniques. Therefore, while rs833069 does not appear to be a robust genetic marker for DR susceptibility in T2DM, its potential influence on treatment outcomes and disease progression warrants further investigation.

#### 3.2.2. *VEGFR2* Polymorphisms

The vascular endothelial growth factor receptor 2 (VEGFR-2), also referred to as kinase insert domain receptor (KDR), is the primary receptor for VEGFA and is essential for angiogenesis and vascular permeability [9,10]. Evidence also indicates that it may participate in lymphangiogenesis through binding to VEGF-C and VEGF-D, though the exact pathways are not yet fully understood [9].

The *VEGFR2* gene is located on chromosome 4q11 [169], divided into 30 exons and 29 introns [170]. *VEGFR2* polymorphisms have emerged as an important area of investigation, given their reported associations with diverse conditions, such as MI among Caucasians with T2DM [71,78] e RD [171,172,173,174].

##### *VEGFR2* rs2071559

The polymorphism rs2071559 (−604T/C) at the −604 located in the promoter region of the *VEGFR2* gene at the binding site for the transcriptional factor E2F (which regulates cell cycle progression), has been shown to suppress transcriptional activity and down-regulate the expression of VEGFR2 [175].

A total of 6 studies in this review evaluated rs2071559 in relation to various diabetic complications. Regarding DR, Yang and collaborators (2014) found a significant association between rs2071559 and DR, reporting that individuals carrying the CC genotype had an increased risk for developing the condition [73]. This suggests that the variant may influence microvascular pathology via altered VEGFR2 expression. In contrast, other studies found no significant associations with DR or T2DM, suggesting that rs2071559 may not play a direct role in these complications in the studied populations. The lack of association observed in these studies may reflect the complex interplay between genetic, environmental, and clinical factors in diabetic complications [44,74,80].

For macrovascular disease, Kariz and colleagues (2014) reported that the CC genotype was significantly more frequent among diabetic patients with MI than those without coronary artery disease (CAD), indicating a 1.6-fold increased risk for MI (OR = 1.6; 95% CI = 1.1–2.1; *p* = 0.022). They also found elevated VEGF serum levels in CC carriers compared to CT or TT genotypes (*p* < 0.01), reinforcing the functional relevance of this promoter variant [71]. These findings support the notion that rs2071559 may influence systemic VEGF levels, indicating a potential functional consequence of this variant in promoting angiogenesis and vascular dysfunction, even if not directly associated with retinopathy outcomes in all studies [80]. Furthermore, another study demonstrated that the C allele was associated with a higher risk of DN, suggesting that rs2071559 may contribute to renal microvascular injury in diabetes [102].

Remarkably, certain cohorts have reported that individuals carrying the CC genotype demonstrate higher circulating levels of VEGF, which aligns with an increased angiogenic drive in specific vascular beds, such as ocular or macrovascular areas. In contrast, within the cutaneous wound microenvironment of individuals with T2DM, research indicates a diminished induction of VEGFA and VEGFR2 signaling, leading to impaired angiogenesis and delayed wound healing. These observations highlight the context-dependent dysregulation of the VEGF axis across various tissues and compartments.

In summary, the rs2071559 variant in *VEGFR2* shows potential associations with multiple diabetic complications, including DR, MI, and DN. However, inconsistencies across studies highlight the importance of replication in diverse populations and further functional validation. Its role may be modulated by complex gene-environment interactions and disease-specific mechanisms of angiogenesis.

##### *VEGFR2* rs2305948

The *VEGFR2* rs2305948 (1192C/T) variant in exon 7 induces an amino acid substitution (Val > Ile) at residue 297, located in the third NH_2_-terminal Ig-like domains within the extracellular region in VEGFR2, which is essential for ligand binding [176]. VEGF–VEGFR2 interactions regulate vascular function, inflammation, angiogenesis, and may influence renal outcomes in conditions such as DN [177].

Abdelghany and collaborators (2021) did not observe any significant association between rs2305948 and susceptibility to DR or T2DM in the Egyptians. However, they noted this variant had previously been linked to age-related macular degeneration (AMD) and response to anti-VEGF therapy in other populations, indicating possible context- and disease-specific roles [90].

Supporting a potential vascular role, other authors found that the C allele of rs2305948 was significantly associated with diabetic nephropathy (DN), further implicating this variant in microvascular complications of diabetes [102].

Taken together, these findings suggest that the rs2305948 polymorphism may contribute to vascular complications in diabetes, particularly DN, potentially through its impact on VEGF receptor function and downstream signaling. However, its role in DR remains unclear, and the observed variability across studies may be influenced by population-specific genetic backgrounds and environmental factors. Further functional and large-scale association studies are necessary to confirm its pathogenic relevance and utility as a biomarker.

#### 3.2.3. *VEGFR1* Polymorphisms

The *VEGFR1(Flt*-1) gene, located on chromosome 13q12 [178], contains at least 30 coding exons, all of which are included in the full-length FLT1 transcript [179]. The *VEGF1* gene encodes both a receptor tyrosine kinase (membrane FLT1 or VEGF-1) and a secreted splice variant (sFLT1 or soluble VEGFR1) which consists of the flt-1 extracellular domain [179]. VEGFR1 binds to several ligands, including VEGFA, VEGFB, and PlGF [7]. VEGFR1 contributes to monocyte and macrophage recruitment during inflammation and tissue repair [7]. The alternative soluble form (soluble VEGFR1, sFlt1) of VEGFR1, a splice variant which acts as an inhibitor for VEGFR activity, it has a higher affinity for VEGFA when compared with VEGFR2, reducing local concentrations of growth factor and limiting VEGFA binding to VEGFR2 [180].

##### *VEGFR1* rs7993418

The rs7993418 is a synonymous T/C SNP located in *VEGFR1* exon 28 that causes a shift in codon usage of tyrosine 1213 [181], the major VEGFR1 autophosphorylation site [182], leading to increased VEGFR1 expression in C allele carriers and downstream VEGFR1 signaling [181]. The CC genotype was associated with decreased serum sVEGFR-1 concentration [183], and conferred protection against DR [90].

In the study of Abdelghany and collaborators (2021), a significant association was observed between the G allele of rs7993418 and susceptibility to T2DM and DR. This suggests that this polymorphism may modulate VEGFR1-mediated signaling pathways involved in the pathogenesis of diabetes-related microvascular complications [90].

Despite being the only study available in this review addressing this SNP, the findings highlight its potential as a genetic biomarker. However, additional studies across diverse populations and functional analyses are needed to confirm its role and mechanism of action in diabetic complications.

#### 3.2.4. *VEGFB* Polymorphisms

The vascular endothelial growth factor B *(VEGFB*) gene, located on chromosome 11q13.1, encodes VEGFB [184]. Inhibition of VEGFB signaling has been shown to reduce ectopic lipid accumulation, enhance peripheral insulin sensitivity and muscle glucose uptake, and preserve islet function. These effects highlight VEGFB as a potential therapeutic target for addressing the underlying pathology of T2DM and its metabolic complications [185].

##### *VEGFB* rs12366035

The rs12366035 C/T variant, which is a synonymous variant located in exon 5 [186], has been linked to circulating VEGFB levels associated with DR [187] and to newly diagnosed T2DM, showing strong correlations with glucose and lipid metabolism as well as first-phase insulin secretion by β-cells. These findings suggest that VEGF-B may contribute to β-cell dysfunction in T2DM [188].

The study by Abdelghany and collaborators (2021) highlights that the T allele is associated with a higher susceptibility to developing T2DM and confers an increased risk for DR progression. However, due to the limited number of studies on each polymorphism, further research is necessary to validate these associations and elucidate the underlying genetic mechanisms, particularly in diverse populations [90].

#### 3.2.5. *VEGFC* Polymorphisms

The *VEGFC* gene is located on chromosome 4q34 [184] and shares a high degree of homology with VEGFA [189]. Although VEGFC can interact with VEGFR2, its main activity is mediated by VEGFR3 [189]. It is recognized as a major regulator of lymphangiogenesis, supporting lymphatic endothelial cell survival, proliferation, and migration via VEGFR3-mediated pathways [190].

##### *VEGFC* rs7664413

The rs7664413 polymorphism is located on intron 5 of the *VEGFC* gene, in a sequence of a putative exonic splicing silencer, suggesting that the rs7664413 SNP might affect VEGFC mRNA [191].

In the only study identified in this review, Abdelghany and collaborators (2021) investigated the rs7664413 variant and reported a significant association between the T allele and increased susceptibility to DR and T2DM. These findings suggest that rs7664413 may act as a genetic risk factor by influencing VEGFC expression and, consequently, vascular function and inflammation [90].

Despite the promising association, the evidence remains limited to a single study, and further research is necessary to validate these findings in different populations and to clarify the underlying mechanisms. Functional studies are also essential to determine whether this variant directly impacts VEGFC expression and contributes to the pathogenesis of diabetic microvascular complications.

#### 3.2.6. Other Polymorphisms

Although the polymorphism rs10738760 is located on chromosome 9p24.2, between the *VLDLR* and *KCNV2* genes (that encode the very low density lipoprotein receptor and the potassium voltage-gated channel subfamily V, member 2, respectively), was significantly associated with serum VEGF levels and explained 5.0% of the VEGF variance [129].

The polymorphism rs10738760 showed no significant link to diabetes or its complications. This polymorphism, located near the *VEGFA* gene, was not associated with T2DM susceptibility or diabetic nephropathy and retinopathy in the French, Danish [68] and Slovenian populations [22,164]. These findings indicate that VEGF-related genetic factors may interact with other molecular pathways, requiring further research to clarify their exact role in diabetes pathophysiology.

### 3.3. Expert Opinion and Future Perspectives

The studies included in this scoping review offer significant insights into the role of *VEGF* gene polymorphisms in T2DM and its related complications; however, several limitations must be addressed to enhance the validity of the conclusions. One notable limitation lies in the heterogeneity of the populations examined. Although a variety of ethnic groups were included, most of the studies were geographically concentrated in regions like China, India, and Egypt. This geographical bias hampers the extrapolation of findings to populations with different genetic backgrounds. Additionally, many studies reported relatively small sample sizes, which could undermine the statistical power necessary for drawing definitive conclusions. In this sense, Figure 5 summarizes the current distribution of evidence regarding VEGFA and its receptor gene polymorphisms in relation to T2DM and its complications across different populations in the revised literature. The figure clearly reveals marked disparities in research coverage, with certain populations—such as East Asian and European—being more frequently studied, particularly in relation to DR, while others, like Latin American and African populations, remain largely underrepresented. This uneven evidence landscape underscores the need for more inclusive and geographically diverse studies, particularly in populations that are currently understudied, to strengthen the generalizability and clinical relevance of genetic association findings.

Additionally, as suggested in some studies [68,164,192,193], the association of VEGF variants with T2DM and microvascular complications appears multifactorial, involving both genetic and environmental influences such as disease duration, glycemic control, and systemic inflammation affecting the axis VEGF-VEGFR in T2DM.

Looking ahead, there is significant potential for translating data on *VEGF* polymorphisms into clinical applications, particularly for risk stratification and personalized medicine in T2DM patients. Identifying genetic markers associated with complications like diabetic retinopathy, nephropathy, and neuropathy could enable early detection and more accurate risk prediction. Individuals carrying specific *VEGF* SNPs associated with a higher risk of complications could be subject to more rigorous monitoring and tailored interventions aimed at halting or delaying the progression of these conditions. However, to progress from genetic associations to clinical applications, further research is essential to elucidate the functional implications of these polymorphisms and their influence on disease progression at the molecular level.

Moreover, translating *VEGF* polymorphism data into preventive strategies must navigate the challenges posed by genetic diversity and environmental influences. While variants such as rs3025039 and rs2010963 demonstrate promising associations with microvascular complications, their application in routine clinical practice is limited by the complex interplay of genetic, lifestyle, and environmental factors. Future investigations should strive to integrate genetic data with additional biomarkers and clinical risk factors, aiming to develop multifactorial models that enhance complication prediction. Such models could inform preventive strategies and individualized therapeutic approaches within clinical settings.

Ultimately, the successful translation of these findings into clinical practice relies on the development of clinically actionable genetic tests. For these tests to be truly beneficial, they must be cost-effective, accurate, and easily accessible for routine screening. Considering the growing importance of genetic testing in personalized medicine, it is crucial to demonstrate the clinical utility of *VEGF* SNPs not only in predicting disease risk but also in informing clinical decision-making. Efforts should be directed toward incorporating genetic testing into clinical guidelines for the management of T2DM and its complications, with the overarching goal of enhancing prevention, facilitating early intervention, and tailoring treatment plans to individual patients.

To our knowledge, this scoping review is the first to comprehensively map polymorphisms across the VEGF signaling pathway in relation to complications of T2DM. By summarizing the most frequently studied variants and their putative links to micro- and macrovascular outcomes, it consolidates a fragmented literature and identifies key gaps, particularly the need for broader ancestral representation and methodological standardization. Consistent with the remit of scoping reviews, we mapped the evidence base without formal quality appraisal or quantitative synthesis; accordingly, we did not establish direct correlations between *VEGF* and *VEGFR* polymorphisms. Rather, this evidence map lays the foundation for a pre-registered systematic review and meta-analysis that we are planning to evaluate pooled associations and potential cross-gene correlations. To strengthen future work, we highlight five priorities: (i) adequately powered, multi-ancestry cohorts with harmonized clinical definitions and consistent covariate adjustment; (ii) standardized reporting and genetic quality control; (iii) a systematic review and meta-analysis using random-effects and trans-ethnic models with sensitivity and small-study bias analyses; (iv) functional and integrative studies of leading variants; and (v) translation-oriented research, including Mendelian randomization, externally validated polygenic risk scores with decision-curve analysis, and pharmacogenetic evaluation of anti-VEGF response. Together, these steps offer a practical roadmap toward more robust, generalizable, and clinically meaningful insights into genetic risk across the VEGF axis.

## 4. Conclusions

This scoping review highlights the significant role of *VEGF* gene polymorphisms and their receptors in the susceptibility to T2DM and its associated microvascular and macrovascular complications. Our analysis of 59 studies demonstrated that certain SNPs, such as rs3025039, rs2010963, rs699947, rs1570360, and rs833061 in the *VEGFA* gene, show variable associations with T2DM and complications.

Based on the descriptive evidence mapping, rs3025039 has more often been reported in association with DR/PDR than with T2DM susceptibility per se; however, findings for T2DM are heterogeneous, and some studies have suggested a higher risk for the TT genotype. Similarly, the rs2010963 SNP has been linked to increased VEGF expression, which could promote vascular permeability and contribute to microvascular complications. On the other hand, rs699947 has demonstrated protective effects in some populations against diabetic foot complications but has also been associated with increased VEGF expression, which could contribute to retinal complications.

Other polymorphisms, such as *VEGFA* rs1570360 and rs833061, have shown potential implications in angiogenesis and vascular dysfunction, which may influence the pathogenesis of diabetic complications. While *VEGFR2* rs2071559 was included in several studies, it did not present a strong association with T2DM or its complications. Additionally, in the reviewed studies, SNPs such as *VEGFA* rs13207351 and *VEGFB* [90] were not significantly correlated with disease risk.

From a clinical perspective, these polymorphisms may serve as potential biomarkers to identify individuals at higher risk for developing microvascular or macrovascular complications, allowing for earlier surveillance and targeted interventions. Furthermore, insights into VEGF-related pathways may inform the development of novel therapeutic strategies, including precision medicine approaches aimed at modulating angiogenesis in susceptible individuals. Integrating genetic information into risk assessment models could ultimately improve individualized management and reduce the burden of diabetes-related complications.

Overall, these findings underscore the complexity of the genetic factors involved in the pathogenesis of T2DM and its complications. Although some SNPs show promising associations, inconsistencies among studies highlight the need for further research with larger and more diverse populations to validate these findings. Future investigations should also focus on functional studies to elucidate the biological mechanisms through which these polymorphisms modulate VEGF expression and contribute to disease progression. Understanding these genetic associations could provide valuable insights for developing personalized medicine approaches to mitigate the risk and severity of diabetic complications.

## Figures and Tables

**Figure 1 biomedicines-13-02242-f001:**
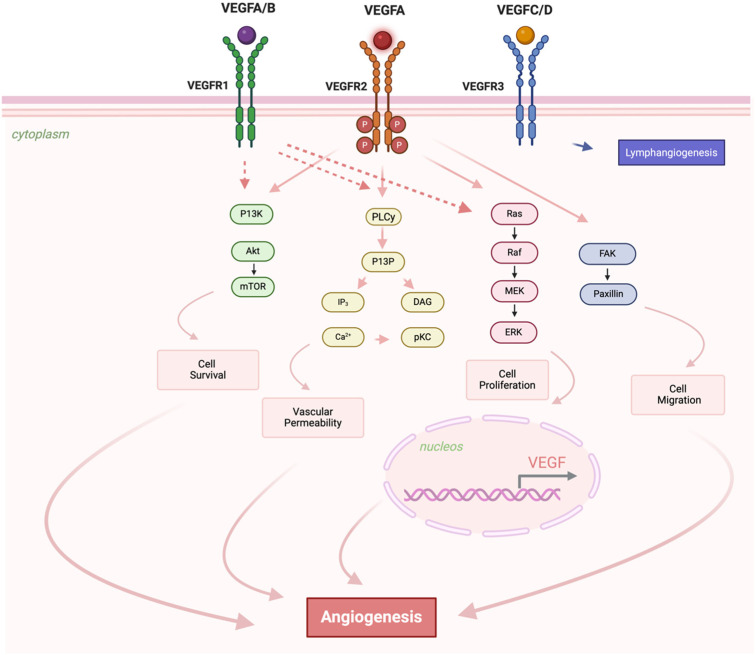
VEGF signaling pathways and their role in angiogenesis and lymphangiogenesis. The figure illustrates the molecular mechanisms initiated by VEGF ligands (VEGFA, VEGFB, VEGFC, and VEGFD) binding to their corresponding receptors (VEGFR1, VEGFR2, and VEGFR3) on endothelial cells. Activation of VEGFR1 primarily triggers the PI3K/Akt/mTOR pathway, promoting cell survival. VEGFR2 activation initiates several downstream cascades, including PLCγ/PIP3/IP3-DAG/Ca^2+^ and Ras/Raf/MEK/ERK pathways, leading to vascular permeability, proliferation, and angiogenesis. VEGFR3 activation by VEGFC/D predominantly regulates lymphangiogenesis via Ras-dependent and FAK pathways, contributing to cell migration and lymphatic vessel formation. The integration of these signaling pathways regulates endothelial cell functions critical for angiogenesis and vascular homeostasis. **Abbreviations:** VEGFA, Vascular Endothelial Growth Factor; VEGFA, Vascular Endothelial Growth Factor A; VEGFB, Vascular Endothelial Growth Factor B; VEGFC, Vascular Endothelial Growth Factor C; VEGFD, Vascular Endothelial Growth Factor D; VEGFR1, Vascular Endothelial Growth Factor Receptor 1; VEGFR2, Vascular Endothelial Growth Factor Receptor 2; VEGFR3, Vascular Endothelial Growth Factor Receptor 3; PI3K, Phosphoinositide 3-Kinase; Akt, Protein Kinase B; mTOR, Mammalian Target of Rapamycin; PLCγ, Phospholipase C gamma; P13P, Phosphatidylinositol 3-Phosphate; IP_3_, Inositol 1,4,5-Trisphosphate; DAG, Diacylglycerol; Ca^2+^, Calcium ion; PKC, Protein Kinase C; Ras, Ras proto-oncogene GTPase; Raf, Rapidly Accelerated Fibrosarcoma kinase; MEK, MAPK/ERK Kinase; ERK, Extracellular signal-Regulated Kinase; FAK, Focal Adhesion Kinase; Paxillin, cytoskeletal adaptor protein. Created with BioRender.com.

**Figure 2 biomedicines-13-02242-f002:**
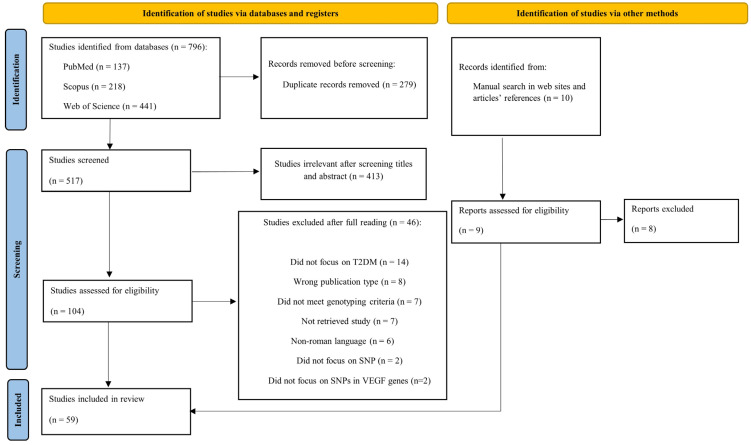
Flowchart of the study selection process.

**Figure 3 biomedicines-13-02242-f003:**
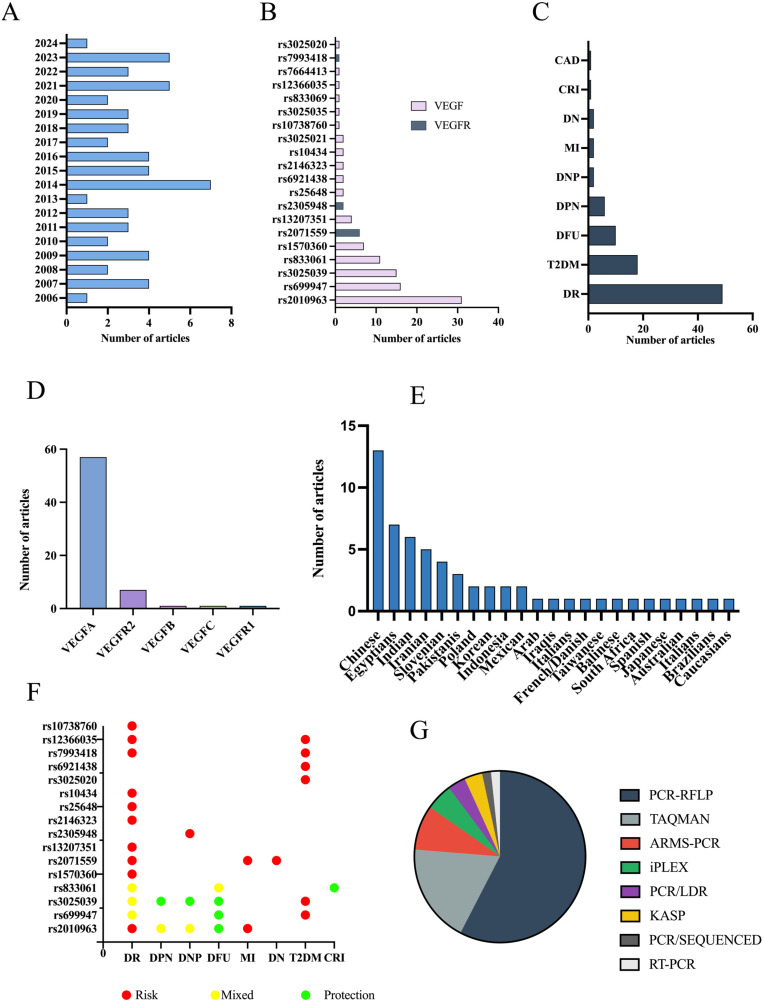
Flow Descriptive statistical analysis of *VEGF* and *VEGFR* gene polymorphism studies in T2DM and related complications. This figure presents the distribution and characteristics of studies included in the review. (**A**) Year of the publication; (**B**) Frequency of individual SNPs evaluated-for clarity, each SNP is labeled by gene family: *VEGF* genes—rs10738760, rs12366035, rs6921438, rs3025020, rs10434, rs25648, rs2146323, rs13207351, rs1570360, rs833061, rs3025039, rs699947, rs2010963; *VEGFR* genes—rs7993418 (*VEGFR1*), rs2305948 (*VEGFR2*), rs2071559 (*VEGFR2*); (**C**) Number of studies per SNP-complication association; (**D**) Distribution of SNPs by gene; (**E**) Associations between SNPs and diabetic complications, with red indicating risk, green indicating protection; (**F**) Frequency of SNPs grouped by study population; (**G**) Distribution of genotyping methodologies used across studies. **Abbreviations:** T2DM: Type 2 Diabetes Mellitus, DR: Diabetic Retinopathy, PDR: Proliferative Diabetic Retinopathy, NPDR: Non-Proliferative Diabetic Retinopathy, DPN: Diabetic Peripheral Neuropathy, DN: Diabetic Nephropathy, DFU: Diabetic Foot Ulcer, VEGFA, VEGFB, VEGFC—Vascular Endothelial Growth Factor isoforms A, B, and C; VEGFR1, VEGFR2—Vascular Endothelial Growth Factor Receptors 1 and 2; PCR-RFLP: Polymerase Chain Reaction–Restriction Fragment Length Polymorphism, TAQMAN: TaqMan Allelic Discrimination Assay, ARMS-PCR: Amplification Refractory Mutation System Polymerase Chain Reaction, iPLEX—iPLEX MassARRAY Genotyping, PCR/LDR: Polymerase Chain Reaction–Ligase Detection Reaction KASP (Kompetitive Allele Specific PCR) e RT-PCR (Reverse Transcription Polymerase Chain Reaction).

**Figure 4 biomedicines-13-02242-f004:**
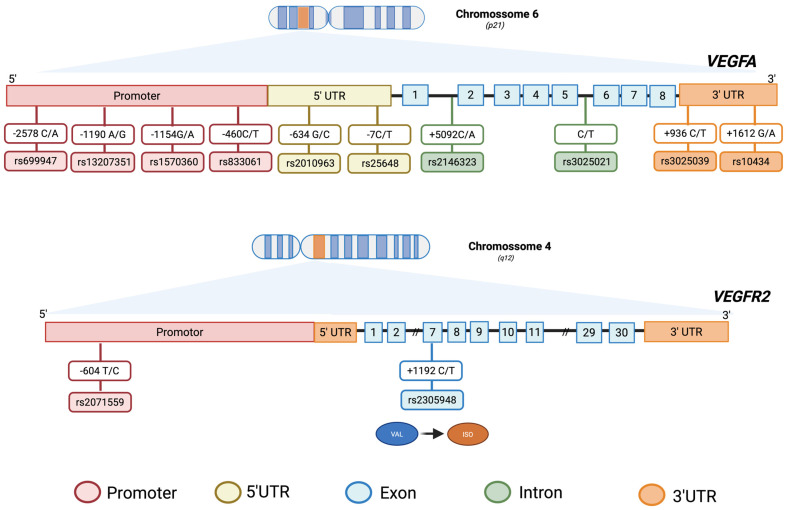
Structure of the *VEGFA* and *VEGFR2* genes and locations of SNPs covered in this review. Schematic representation of the *VEGFA* gene on chromosome 6 and the *VEGFR2* gene on chromosome 4, indicating the positions of selected SNPs relative to gene structure. They are mapped according to their genomic locations in the promoter, untranslated regions (5′ UTR and 3′ UTR), introns, or exons. In *VEGFA*, the rs3025039 and rs1034 are located in the 3′ UTR, while rs699947, rs2010963, rs833061, rs13207351, rs2146323, and rs1570360 are found in the promoter region or 5′ UTR. The rs2146323 and rs3025021 are located in intronic regions (intron 2 e intron 7). In *VEGFR2*, rs2071559 is located in the promoter region, and rs2305948 is in exon 7, where it results in a Valine (VAL) to Isoleucine (ISO) amino acid substitution. Illustration created with BioRender.com.

**Figure 5 biomedicines-13-02242-f005:**
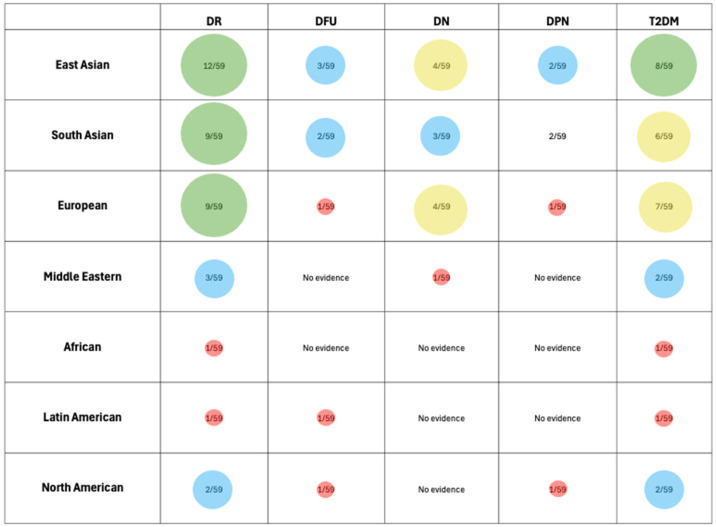
Number of studies investigating the association between *VEGFA* gene polymorphisms (and its receptors) and T2DM complications, stratified by population and complication type. The complications include diabetic retinopathy (DR), diabetic foot ulcer (DFU), diabetic nephropathy (DN), and diabetic peripheral neuropathy (DPN). ‘T2DM’s column indicates studies of patients with T2DM without recorded micro- or macrovascular complications. Complication columns (DR, DN, DPN, DFU) indicate studies that explicitly analyzed those endpoints in T2DM. Color significance: Green circle—strong evidence (≥8 studies); yellow circle—moderate evidence (4–7 studies); blue circle—low-to-moderate evidence (2–3 studies); red circle—low evidence (1 study).

**Table 1 biomedicines-13-02242-t001:** Search strategies for PubMed, Scopus, and Web of Science articles.

Database	Queries
PubMed	((((((Polymorphism, Single Nucleotide[MeSH Terms]) OR (Polymorphism*[Title/Abstract])) OR (“SINGLE NUCLEOTIDE POLYMORPHISM”[Title/Abstract])) OR (SNP[Title/Abstract])) AND (((((((“Vascular Endothelial Growth Factor A”[MeSH Terms]) OR (“Vascular Endothelial Growth Factor A”[Title/Abstract])) OR (“Vascular Endothelial Growth Factor”[Title/Abstract])) OR (“Vascular Endothelial Growth Factor RECEPTOR”[Title/Abstract])) OR (VEGF[Title/Abstract])) OR (VEGF-A[Title/Abstract])) OR (“Permeability Factor, Vascular”[Title/Abstract]))) AND ((((((“Diabetes Mellitus”[MeSH Terms]) OR (“Diabetes complications”[MeSH Terms])) OR (“DIABETES MELLITUS”[Title/Abstract])) OR (DM[Title/Abstract])) OR (Diabetic[Title/Abstract])) OR (“Diabetic complications”[Title/Abstract]))) AND ((((((((((((neuropathy[Title/Abstract]) OR (Painful[Title/Abstract])) OR (“peripheral neuropathy”[Title/Abstract])) OR (polyneuropathy[Title/Abstract])) OR (“neuropathic pain”[Title/Abstract])) OR (retinopathy[Title/Abstract])) OR (nephropathy[Title/Abstract])) OR (“Microvascular complications”[Title/Abstract])) OR (“Macrovascular complications”[Title/Abstract])) OR (“Cerebrovascular Accident”[Title/Abstract])) OR (Stroke[Title/Abstract])) OR (“Vascular periphery”[Title/Abstract]))
Scopus	((TITLE-ABS-KEY (“polymorphism, single nucleotide”) OR TITLE-ABS-KEY (polymorphism*) OR TITLE-ABS-KEY (“single nucleotide polymorphism”) OR TITLE-ABS-KEY (SNP))) AND ((TITLE-ABS-KEY (“vascular endothelial growth factor a”) OR TITLE-ABS-KEY (“vascular endothelial growth factor”) OR TITLE-ABS-KEY (“vascular endothelial growth factor receptor”) OR TITLE-ABS-KEY (VEGF) OR TITLE-ABS-KEY (VEGF-a) OR TITLE-ABS-KEY (“permeability factor, vascular”))) AND ((TITLE-ABS-KEY (“diabetes mellitus”) OR TITLE-ABS-KEY (“diabetes complications”) OR TITLE-ABS-KEY (dm) OR TITLE-ABS-KEY (diabetic))) AND ((TITLE-ABS-KEY (neuropathy) OR TITLE-ABS-KEY (painful) OR TITLE-ABS-KEY (“peripheral neuropathy”) OR TITLE-ABS-KEY (polyneuropathy) OR TITLE-ABS-KEY (“neuropathic pain”) OR TITLE-ABS-KEY (retinopathy) OR TITLE-ABS-KEY (nephropathy) OR TITLE-ABS-KEY (“microvascular complications”) OR TITLE-ABS-KEY (“macrovascular complications”) OR TITLE-ABS-KEY (“cerebrovascular accident”) OR TITLE-ABS-KEY (stroke) OR TITLE-ABS-KEY (“vascular periphery”)))
Web of Science	“Polymorphism, Single Nucleotide” (Topic) or Polymorphism* (Topic) or “SINGLE NUCLEOTIDE POLYMORPHISM” (Topic) or SNP (Topic) AND TS = (“Vascular Endothelial Growth Factor A”) OR TS = (“Vascular Endothelial Growth Factor”) OR TS = (“Vascular Endothelial Growth Factor RECEPTOR”) OR TS = (VEGF) OR ALL = (VEGF-A) OR TS = (“Permeability Factor, Vascular”) AND TS = (“Diabetes Mellitus”) OR TS = (“Diabetes complications”) OR TS = (DM) OR TS = (Diabetic) AND TS = (neuropathy) OR TS = (Painful) OR TS = (“peripheral neuropathy”) OR TS = (polyneuropathy) OR ALL = (“neuropathic pain”) OR TS = (retinopathy) OR TS = (nephropathy) OR TS = (“Microvascular complications”) OR TS = (“Macrovascular complications”) OR TS = (“Cerebrovascular Accident”) OR TS = (Stroke) OR TS = (“Vascular periphery”)

Terms applied to the search strategies of this review.

**Table 2 biomedicines-13-02242-t002:** General characteristics of the selected studies.

Reference	Ethnicity	Gene	SNP ID	Methods	Ctrl (*n*)	Cases (*n*)	Result
Suganthalakshmi et al., 2006 [50]	Indian	* VEGFA *	rs2010963 (−634G/C) rs25648 (−7C/T)	PCR-RFLP	90	120	Associated with DR. The heterozygous genotypes of rs25648 and rs201093 were 4.17 (95% CI: 1.90–9.18, *p* = 0.0001 and 2.33 (95% CI: 1.24–4.36, *p* = 0.008), respectively. Significantly higher in the DR group when compared with control.
Buraczynska et al., 2007 [34]	Poland	* VEGFA *	rs2010963 (−634G/C)	PCR-RFLP	493	426	No association was found between genotype and DNP or DR in the patient population.
Errera et al., 2007 [51]	Brazilian	* VEGFA *	rs2010963 (−634G/C)	PCR-RFLP	334	167	The CC genotype is an independent risk factor for PDR in T2DM of European ancestry OR = 1.9; IC 95%: 1.01–3.79; *p* = 0.04
Petrovic et al., 2007 [52]	Slovenian	* VEGFA *	rs2010963 (−634G/C)	PCR-RFLP	228	143	The CC genotype may be a risk factor for MI in long-term T2DM patients (OR = 2.1; 95% CI = 1.1–3.9; *p* = 0.019)
Szaflik et al., 2007 [53]	Poland	* VEGFA *	rs2010963 (−634G/C) rs833061 (−460T/C)	PCR-RFLP	61	72	The Allele C is associated with increased *VEGF* gene promoter activity, and the GC genotype predictive factor for the development of DR in rs2010963. No association with rs833061.
Petrovic et al., 2008 [54]	Slovenian	* VEGFA *	rs2010963 (−634G/C)	PCR-RFLP	143	206	the *VEGF* –634 C/G polymorphism failed to contribute to the genetic susceptibility to PDR.
Uthra et al., 2008 [55]	Indian	* VEGFA *	rs2010963 (−634G/C) rs3025039 (+936C/T)	PCR/Sequenced	82	131	No association for DR in rs3025039. The GC genotype in rs2010963 increased risk for DR in patients with microalbuminuria (OR = 8.9; 95% CI: 1.4, 58.3).
Abhary et al., 2009 [56]	Australian	* VEGFA *	rs3025021 (C/T) rs10434 (+1612G/A)	iPLEX	187	139	The C allele of rs3025021 (*p* = 0.002; OR, 3.8; 95% CI, 1.5–10.0) and the G allele of rs10434 (*p* = 0.002; OR, 2.6; 95% CI, 1.3–5.3) significantly associated with blinding DR.
Kim et al., 2009 [57]	Korean	* VEGFA *	rs3025039 (+936 C/T)	PCR-RFLP	526	398	Diabetics with DR showed a higher frequency of TT genotype and T Allele, it is associated with higher levels of VEGF and DR
Nakamura et al., 2009 [58]	Japanese	*VEGFA*	rs2010963 (−634G/C) rs699947 (−2578C/A)	PCR-RFLP	292	177	The AA genotype of rs699947 is associated with PDR, the risk is 7.7 (95%, CI: 1.8–30.9). No association in rs2010963.
Tiwari et al., 2009 [59]	Indian	*VEGFA*	rs833061 (−460T/C) rs2010963 (−634G/C)	PCR-RFLP	224	194	Significant association of rs833061 with CRI (*p* < 0.05) The CT genotype, OR = 2.23 (1.87–4.18). No association in rs2010963
Chun et al., 2010 [60]	Korean	*VEGFA*	rs699947 (−2578C/A) rs1570360 (−1154G/A) rs2010963 (−634G/C)	TAQMAN	260	387	The A allele at rs699947 is significant association with DR.
Yang et al., 2010 [61]	Chinese	* VEGFA *	rs2010963 (−634G/C)	TAQMAN	96	285	The rs2010963 SNP is not associated with T2DM but increases the risk of DR.
Feghhi et al., 2011 [62]	Iranian	* VEGFA *	rs2010963 (−634G/C)	PCR-RFLP	279	119	The GG genotype associated with the risk of PDR (OR:1.87, 95%, CI (1.034–3.383).
Amoli et al., 2011 [63]	Iranian	* VEGFA *	rs25648 (−7C/T) rs699947 (−2578C/A)	ARMS-PCR	98	488	The genotype AA of rs699947 was significantly reduced in DFU cases, conferring a protective effect.
Yang et al., 2011 [64]	Chinese	*VEGFA*	rs699947 (−2578C/A) rs833061 (−460T/C) rs13207351 (−1190A/G) rs2010963 (−634G/C) rs2146323 (+5092C/A) rs3025039 (+936C/T)	iPLEX	139	129	A significant association of DR was observed with the AA genotype of rs699947 (odds ratio (OR) = 3.54, 95% confidence interval (CI): 1.12–11.19), the CC genotype of rs833061 (OR = 3.72, 95% CI: 1.17–11.85) and the AA genotype of rs13207351 (OR = 3.76, 95% CI: 1.21–11.71.
Bleda et al., 2012 [65]	Spanish	*VEGFA*	rs2010963 (−634G/C) rs699947 (−2578C/A)	TAQMAN	14	26	The CC genotype of rs2010963 was increased in PAD, while the CA genotype of rs699947 was more prevalent in DR (*p* = 0.016 and *p* = 0.002, respectively).
Nikzamir et al., 2012 [66]	Iranian	*VEGFA*	rs2010963 (+405G/C)	PCR-RFLP	235	255	The GG genotype is independently associated with development of DNP [*p* = 0.014, OR = 1.771, 95% confidence interval (CI) = 1.124–2.790]. The G allele was not associated with albuminuria.
Paine et al., 2012 [67]	Indian	*VEGFA*	rs833061 (−460T/C)	PCR-RFLP	240	253	The CC genotype significantly associated with PDR (OR [95% CI]3.66(1.35–11.4))
Bonnefond et al., 2013 [68]	French Danish	*VEGFA*	rs6921438 (A/G) rs10738760 (A/G)	TAQMAN	3875 3561	6920 2623	Association with the G allele of rs692143 and T2D in the French population
El-Shazly et al., 2014 [69]	Egyptians	* VEGFA *	rs2010963 (−634G/C)	PCR-RFLP	180	212	The *CC* genotype is a risk factor for DR. (*p* < 0.001)
Fan et al., 2014 [70]	Chinese	* VEGFA *	rs699947 (−2578C/A) rs1570360 (−1154G/A) rs833061 (−460T/C) rs2010963 (−634G/C) rs3025039 (+936C/T)	PCR-RFLP	668	372	No association between the SNP’s and DR. Patients with DR have higher VEGF level and rs699947 and rs2010963 may be important factor for serum VEGF levels.
Kariz et al., 2014 [71]	Slovenian	* VEGFR2 *	rs2071559 (−604 T/C) rs2305948 (+1192C/T)	PCR-RFLP	850	171	The CC genotype of the rs2071559 is a risk factor for MI (OR = 1.6; 95% CI = 1.1–2.1; *p* = 0.022).
Pirie et al., 2014 [72]	South African	* VEGFA *	rs2010963 (−634G/C)	PCR-RFLP	171	117	No significant association found between SNPs and DR in South African population
Yang et al., 2014 [73]	Chinese	* VEGFA * * VEGFR2 *	rs699947 (−2578C/A) rs833061 (−460T/C) rs13207351 (−1190A/G) rs2146323 (+5092C/A) rs2071559 (−604T/C)	iPLEX	284	216	DR showed significant associations with *VEGF* SNPs—rs699947 (*p* < 0.001), rs833061 (*p* = 0.001), rs13207351 (*p* < 0.001), and rs2146323 (*p* = 0.006)—as well as with one variant in the *VEGFR2* gene, rs2071559 (*p* = 0.034)
Yuan et al., 2014 [74]	Chinese	* VEGFA *	rs2010963 (−634G/C) rs833061 (−460T/C)	PCR/LDR	134	144	The rs833061 was correlated with NPDR, and C allele was associated with lower NPDR risk than T allele. The rs2010963 not correlated with NPDR or PDR.
Zhang et al., 2014 [45]	Chinese	* VEGFA *	rs3025039 (+936 C/T)	PCR-RFLP	240	184	Allele C of rs3025039 may be a genetic marker susceptible to DPN, while allele T may be a protective marker of DPN.
Choudhury et al., 2015 [44]	Indian	* VEGFA *	rs2010963 (−634G/C) rs3025039 (+936C/T) rs1570360 (−1154G/A) rs2071559 (−604T/C)	TAQMAN	95	102	The rs2010963 C allele and rs3025039 T allele might be associated with PDR occurrence and, in turn, regulate VEGF expression among PDR subjects.
Ghisleni et al., 2015 [75]	Brazilians	* VEGFA *	rs3025039 (+936 C/T)	PCR-RFLP	104	98	The rs3025039 are not correlated with the risk of developing T2DM or neuropathic signs and symptoms.
Moradzadegan et al., 2015 [76]	Iranian	* VEGFA *	rs2010963 (−634G/C)	PCR-RFLP	369	141	The G allele of rs2010963 can be an important independent risk factor for susceptibility of CAD in T2DM patients OR (1.75) (*p* = 0.024).
Porojan et al., 2015 [41]	Caucasians	* VEGFA *	rs3025039 (+936 C/T)	PCR-RFLP	208	200	The rs3025039 revealed an increased risk of T2DM, is highly associated with DR; therefore, this genetic variant is confirmed to be an independent genetic risk factor for NPDR.
Chen et al., 2016 [77]	Taiwanese	* VEGFA *	rs1570360 (−1154G/A) rs2010963 (−634G/C)	TAQMAN	31	53	The rs2010963 is an important genetic marker for DR. The experiments provide a direct association between allele C of rs2010963 and serum levels of VEGFA.
Fattah et al., 2016 [78]	Egyptians	* VEGFA *	rs699947 (−2578C/A) rs10434 (+1612G/A)	RT-PCR	41	41	The rs699947 or rs10434 polymorphism was not associated with DR in Egyptian patients.
Kamal et al., 2016 [79]	Egyptians	* VEGFA *	rs2010963 (−634G/C)	PCR-RFLP	61	61	Patients carrying allele C have a higher risk of PDR development, so rs2010963 could be used as a predictive marker for PDR in diabetic patients.
Merlo et al., 2016 [80]	Slovenian	* VEGFA * * VEGFR2 *	rs2010963 (−634G/C) rs2071559 (−604T/C)	KASP	200	595	There were no statistically significant differences in the *VEGF* rs2010963 and KDR rs2071559 genotype distribution frequencies between T2DM patients and controls.
Gonzales-Salinas et al., 2017 [81]	Mexican	* VEGFA *	rs2010963 (−634G/C) rs3025021 (C/T) rs3025035 (C/T)	TAQMAN	71	71	None of the polymorphisms studied were significantly associated with PDR.
Zhuang et al., 2017 [82]	Chinese	* VEGFA *	rs833061 (−460T/C) rs1570360 (−1154G/A)	PCR-RFLP	188	209	The rs833061 is associated with DFU.
Barus et al., 2018 [83]	Indonesian	* VEGFA *	rs3025039 (+936C/T)	PCR-RFLP	83	69	The CT + TT genotype is associated with a protective effect against DPN (OR = 0.35; *p* = 0.01). There was also a significant association with VEGFA plasma level and duration of diabetes diagnosis.
Xiaolei Li, 2018 [84]	Chinese	* VEGFA *	rs2010963 (−634G/C)	PCR-RFLP	108	121	The CC genotype and C allele of rs2010963 were less common in DFU than in T2DM (OR = 0.36 and 0.63, respectively), but CC carriers showed higher VEGF levels (*p* = 0.007)
Li et al., 2018 [85]	Chinese	* VEGFA *	rs699947 (−2578C/A) rs13207351 (−1190A/G)	PCR-RFLP	103	185	Association of rs699947 with the occurrence of DFU. The minor A allele might reduce the susceptibility to DFU, while no significant association was detected for rs13207351.
Arredondo-García et al., 2019 [86]	Mexican	*VEGFA*	rs3025039 (+936C/T)	PCR-RFLP	128	90	The rs3025039 tended to be a risk factor for the development of DPN, and CT genotype showed a protective effect (OR = 0.52; 95% CI = 0.300–0.90; *p* = 0.019).
Dahlan et al., 2019 [87]	Indonesia	*VEGFA*	rs833061 (−460T/C) rs2010963 (−634G/C)	PCR-RFLP	101	96	There was no significant relationship between SNPs with DFU. G and T alleles have a potential protective factor against the occurrence of DFU. (OR 0.90, 95% CI; 0.59 to 1.37 and *p* = 0.641).
Luo et al., 2019 [36]	Chinese	*VEGFA*	rs2010963 (−634G/C) rs699947 (−2578C/A)	PCR-RFLP	650	580	The rs2010963 and rs699947 may increase the risk of developing DPN. ([OR] = 1.15, confidence interval [95% CI]: 1.03–1.30)
Yari et al., 2020 [88]	Iranian	*VEGFA*	rs3025039 (+936C/T)	PCR-RFLP	80	80	No association between rs3025039 and T2DM.
Khan et al., 2020 [89]	Pakistanis	* VEGFA *	rs833061 (−460T/C) rs13207351 (−1190A/G) rs1570360 (−1154G/A) rs2010963 (−634G/C)	PCR-RFLP	348	1126	There was no association between SNPs and T2DM; The rs13207351 rs13207351was associated with NPDR [OR = 1.97 (95% CI 1.28–3.03, *p* = 9.0 × 10^−3^)].
Abdelghany et al., 2021 [90]	Egyptians	*VEGFA* *VEGFR2* *VEGFB* *VEGFC* *VEGFR1*	rs833069 (+450T/C) rs2305948 (+1192C/T) rs12366035 (C/T) rs7664413 (C/T) rs7993418 (A/G)	TAQMAN	110	125	This study revealed a significant association between the T allele of rs12366035 and rs7664413, and the AG genotype of rs7993418 and T2DM/DR susceptibility.
Elfaki et al., 2021 [40]	Arab	*VEGFA*	rs699947 (−2578C/A)	ARMS-PCR	126	122	The results showed that the CA genotype of the *VEGF* rs699947 was associated with T2DM with OR =2.01, *p*-value = 0.011.
Jin et al., 2021 [91]	Chinese	* VEGFA *	rs2010963 (−634G/C)	KASP	386	316	Ther s2010963 in the *VEGFA* gene are related to the risk of PDR. The CG genotypes of rs2010963 were associated with a decreased risk of PDR (the OR was 0.588, with a 95% CI ranging from 0.366 to 0.946).
Imbaby et al., 2021 [92]	Egyptians	* VEGFA *	rs3025039 (+936 C/T)	PCR-RFLP	40	50	The rs3025039 may be associated with T2DM. However, there is no association with DPN.
Wijaya et al., 2021 [93]	Balinese	*VEGFA*	rs699947 (−2578C/A)	PCR-RFLP	35	33	The rs699947 as a risk factor of DR in patients with T2DM (OR = 13.05; 95% CI = 2.69–63.18; *p* = 0.001).
Mohamed et al., 2022 [94]	Egyptian	*VEGFA*	rs3025039 (+936C/T)	PCR-RFLP	72	72	The rs3025039 genetic variants were not associated with the PDR progression.
Omar et al., 2022 [95]	Egyptian	*VEGFA*	rs3025020 (−583C/T) rs3025039 (+936C/T)	TAQMAN	26	26	The T allele of rs3025020 (OR = 2.67; *p* = 0.04) and both the CT genotype and T allele of rs3025039 (OR = 4.08 and 4.02; *p* = 0.01 and 0.004) were more common in T2DM patients with mixed complications than in controls.
Singh et al., 2022 [96]	Indian	* VEGFA *	rs699947 (−2578C/A)	PCR-RFLP	51	55	No association with DR.
Alnaji et al., 2023 [97]	Iraqis	* VEGFA *	rs2010963 (−634G/C) rs699947 (−2578C/A)	ARMS-PCR	36	134	The rs2010963 GG genotype showed a significant link with DR (OR = 10.29; *p* = 0.004)
Del Cuore et al., 2023 [98]	Italians	* VEGFA *	rs699947 (−2578C/A) rs3025039 (+936C/T)	TAQMAN	20	90	The CC genotype of rs699947 is associated with DFU.
Jehanzeb et al., 2023 [99]	Pakistanis	* VEGFA *	rs833061 (−460T/C)	ARMS-PCR	184	180	Significant association of rs833061 SNP with DR on T2DM
Quayyum et al., 2023 [100]	Pakistanis	* VEGFA *	rs699947 (−2578C/A) rs1570360 (−1154G/A)	ARMS-PCR	150	300	There was a strong association of rs699947 SNP with PDR in T2DM.
Yuan et al., 2023 [101]	Chinese	*VEGFR2*	rs2071559 (−604T/C)	PCR/LDR	114	123	No associations between the rs2071559 SNP and DR or PDR.
Nussdorfer et al., 2024 [102]	Chinese	*VEGFA* *VEGFR2* *VEGFR2*	rs6921438 (G/A) rs2071559 (−604T/C) rs2305948 (+1192C/T)	TAQMAN	553	344	In Slovenians with T2DM, rs2071559 C (*VEGFR2*) increased DN risk, whereas rs6921438 G (*VEGFA*) was protective

**Abbreviations:** DR—Diabetic Retinopathy; NPDR—Non-Proliferative Diabetic Retinopathy; PDR—Proliferative Diabetic Retinopathy; DPN—Diabetic Peripheral Neuropathy; DFU—Diabetic Foot Ulcer; DN—Diabetic Nephropathy; CAD—Coronary Artery Disease; MI—Myocardial Infarction; T2DM—Type 2 Diabetes Mellitus; VEGF—Vascular Endothelial Growth Factor; VEGFA, VEGFB, VEGFC—VEGF isoforms A, B, C; VEGFR1, VEGFR2—Vascular Endothelial Growth Factor Receptors 1 and 2; KDR—Kinase Insert Domain Receptor (synonym for VEGFR2); SNP—Single Nucleotide Polymorphism; UTR—Untranslated Region; PCR-RFLP—Polymerase Chain Reaction–Restriction Fragment Length Polymorphism; ARMS-PCR—Amplification Refractory Mutation System–Polymerase Chain Reaction; TAQMAN—TaqMan Allelic Discrimination Assay; PCR/LDR—Polymerase Chain Reaction–Ligase Detection Reaction; KASP—Kompetitive Allele Specific PCR; RT-PCR—Reverse Transcription Polymerase Chain Reaction; iPLEX—iPLEX MassARRAY Genotyping (based on MALDI-TOF mass spectrometry); PCR/Sequenced—PCR followed by Direct Sequencing; CI—Confidence Interval; OR—Odds Ratio; *p*—*p*-value (probability value used in statistical hypothesis testing).

**Table 3 biomedicines-13-02242-t003:** Summary of *VEGF* SNPs most frequently studied and their Associations.

SNP ID	Gene	Region	Number of Studies	Associated Complications	Association Pattern
rs2010963	*VEGFA*	5′ UTR	31	DR (PDR, NPDR), MI, DFU, DNP, DPN	C allele = risk for PDR, DR, MI; protective for DFU; mixed for DNP/DPN
rs699947	*VEGFA*	Promoter	16	DR, DFU, T2DM/dyslipidemia	The A allele = protective for DFU; risk or no association for DR; CA genotype = altered lipids and ↑ CV risk-C allele lead to ↑ VEGF
rs3025039	*VEGFA*	3′ UTR	15	DR, DPN, DNP, T2DM, DFU	T allele = protective for DR, DPN, DFU; C allele = risk for DPN and T2DM/DR; T allele and CT genotype = risk for PDR
rs833061	*VEGFA*	Promoter	11	DR (PDR, NPDR), DFU, CRI	C allele lead to ↑ VEGF C allele = increased DR, protective for NPDR, mixed for DFU; CT genotype = risk for CRI
rs1570360	*VEGFA*	Promoter	7	DR(PDR), DFU	AA genotype = risk for PDR. DFU no association
rs2071559	*VEGFR2*	Promoter	6	DR, DN, MI	CC genotype = risk for DR, MI, and DN;
rs13207351	*VEGFA*	Promoter	4	DR, DFU	AA genotype = risk for DR. No association with T2DM or DFU
rs2305948	*VEGFR2*	Exon 7	3	DN	C allele = risk for DN
rs2146323	*VEGFA*	Intron 2	2	DR	AA genotype = risk for DR
rs25648	*VEGFA*	5′ UTR	2	DR, DFU	Heterozygous = risk for DR; no association with DFU
rs10434	*VEGFA*	3′ UTR	2	DR	G allele = risk for DR
rs3025021	*VEGFA*	Intron 6	2	DR(PDR)	C allele = risk for DR/PDR
rs3025020	*VEGFA*	Intron 6	1	T2DM	T allele = risk for T2DM
rs3025035	*VEGFA*	Intron 7	1	DR	No association
rs6921438	*VEGFA*	Downstream	1	T2DM/DN	G allele = protective for T2DM/DN
rs833069	*VEGFA*	Intron 2	1	DR	No association
rs7993418	*VEGFR1*	Exon 28	1	T2DM, DR	AG genotype = risk for T2DM and DR
rs12366035	*VEGFB*	Exon 5	1	T2DM, DR	T allele = Risk for T2DM and DR
rs7664413	*VEGFC*	Intron 5	1	T2DM, DR	T allele = Risk for T2D and DR
rs10738760	*VEGF-related*	Intergenic (*VEGF*)	1	DR	G allele = Risk for DR

The upward arrow (↑) denotes an increase. **Abbreviations:** DR—Diabetic Retinopathy; PDR—Proliferative Diabetic Retinopathy; NPDR—Non-Proliferative Diabetic Retinopathy; DPN—Diabetic Peripheral Neuropathy; DFU—Diabetic Foot Ulcer; DN—Diabetic Nephropathy; T2DM—Type 2 Diabetes Mellitus; CAD—Coronary Artery Disease; MI—Myocardial Infarction; CRI—Chronic Renal Insufficiency; VEGF—Vascular Endothelial Growth Factor; VEGFA, VEGFB, VEGFC—VEGF isoforms A, B, and C; VEGFR1, VEGFR2—Vascular Endothelial Growth Factor Receptors 1 and 2; UTR—Untranslated Region; SNP—Single Nucleotide Polymorphism.

## Data Availability

The authors confirm that the data supporting the findings of this study are available within the article.

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
