# Peer review of "Polymorphisms in *VEGF* Signaling Pathway Genes and Their Potential Impact on Type 2 Diabetes Mellitus and Associated Complications: A Scoping Review"

_biomedicines, 2025, doi:10.3390/biomedicines13092242_

Round 1
Reviewer 1 Report
Comments and Suggestions for Authors
some comments are attached

Author Response
# REVIEWER 1
The work shows an extensive review of VEGF and VEGFR gene polymorphism studies in diabetes type 2 and related complications. Although not a clear conclusion could be obtained, the study identify several SNP that could serve as a valuable biomarkers for possible risk detection.
Answer: We appreciate the Reviewer’s positive evaluation of our manuscript. Your observations contributed significantly to improving the scientific clarity, precision, and robustness of the final version of our work. All suggestions were carefully considered and addressed either directly in the revised manuscript or in the specific responses below. The changes made to the text are clearly highlighted in green in the revised version. They are annotated with indications of the respective page and line numbers for transparency and ease of verification.
Once again, we thank you for your valuable contribution to refining our study.
I have some comments:
- Line 246, “a noticeable concentration of studies in recent years.”, please indicate the date; I do not find that in the figure.
Answer: Thank you for your insightful observation. We agree that the original statement lacked clarity. We have since clarified the publication years with greater precision. Among the studies included, five were published in 2021 and another five in 2023, with 2014 also witnessing a notable concentration of studies in a single year. Importantly, between 2021 and 2023, a total of 13 studies were published and included in this review, reflecting a recent surge of interest in this research area. In response to your suggestion, we have revised the Results section to explicitly present this information. The modifications are highlighted in green (Page 15, line 250 to 251).
- Line 426, “In European ancestry populations, results have been mixed. Errera et al. (2007) identified the CC genotype as an independent risk factor for PDR in Brazilian patients…” please correct.
Answer: Thank you for pointing out this imprecision. We corrected the description of the study by Errera et al. (2007). The original cohort consisted of Brazilian patients of European ancestry, and we have revised the text to reflect this accurately, avoiding confusion between geographic location and ancestry.
The following sentence was included in the revised version of the manuscript:
“In European ancestry populations, results have been mixed. Errera et al. (2007) identified the CC genotype as an independent risk factor for PDR in Brazilian patients of European ancestry (OR=1.9; 95% CI: 1.01–3.79; p=0.04…”
The modifications are highlighted in green (Page 22, line 445 to 447).
- Did you found a correlation between VEGF and VEGFR’s SNPs?
Answer: Thank you for your insightful question. As this study was conducted as a scoping review, our primary objective was to map and summarize the existing evidence on VEGF and VEGFR polymorphisms associated with T2DM and its complications. By nature, a scoping review does not involve a formal risk of bias assessment, quality appraisal, or quantitative synthesis (such as meta-analysis), and therefore, it does not seek to establish direct correlations or comparative effects between VEGF and VEGFR SNPs. Rather, our aim was to provide a comprehensive overview of the current landscape, identifying the most frequently investigated variants, the diversity of populations studied, and the heterogeneity of reported associations.
We appreciate your comment, as it emphasizes a significant direction for future research. Indeed, in light of the evidence mapped in this scoping review, we are in the process of developing a proposal for a systematic review and meta-analysis specifically designed to assess pooled associations and potential correlations between VEGF and VEGFR polymorphisms and diabetic outcomes. We included a brief mention about this topic in the revised document, which is highlighted in green (page 36, line 1065 to 1069).
- In figure 3F, to better appreciate the results, please indicate whether SNP belongs to VEGF or VEGFR.
Answer: We appreciate your valuable suggestion. To enhance clarity, we have revised Figure 3F to explicitly categorize each SNP as belonging to the VEGFA or VEGFR genes. Among the 16 SNPs identified in our review, 13 are located within VEGF genes (VEGFA and its related family members), while 3 are found in VEGFR genes. The distribution is as follows:
VEGF genes (n = 13): rs10738760, rs12366035, rs6921438, rs3025020, rs10434, rs25648, rs2146323, rs13207351, rs1570360, rs833061, rs3025039, rs699947, rs2010963.
VEGFR genes (n = 3): rs7993418 (VEGFR1), rs2305948 (VEGFR2), rs2071559 (VEGFR2).
We have updated both the text and the figure to clearly indicate whether each SNP is related to VEGF or VEGFR, thereby improving the interpretation of the distribution of polymorphisms across these gene families. All modifications are highlighted in green (Page 15, line 257 to 260, and Page 17, line 283 to 286).
- In supplementary table “(studies not included), Wrong document type”, what does it mean?
Answer: Thank you for the question. In Supplementary Table S1, the label “Wrong document type” indicates records that, at full-text screening, proved to be non-eligible publication types according to our pre-specified criteria (e.g., reviews, editorials, books/chapters, and conference abstracts) rather than primary human genotyping studies. This is consistent with our Eligibility Criteria in the Methods. We have clarified this in the Supplementary Table caption and harmonized the wording to “Wrong publication type (review/editorial/book chapter/conference abstract)” for clarity.
- Could the authors propose some future strategy to improve the results? The manuscript is valuable in the context of genetic risk assessment. The authors presented the results and documented the findings of their analyses in a satisfactory manner.
Answer: Thank you for the positive evaluation and for encouraging us to articulate strategies for enhancing future outcomes. Building on this scoping review, we identify several opportunities to strengthen the evidence surrounding genetic risk within the VEGF pathway:
Expansion of Cohorts: Establish larger, multi-ancestry cohorts with harmonized phenotypes and ensure consistent covariate adjustment for factors such as age, sex, diabetes duration, HbA1c, blood pressure, lipid levels, smoking, and ancestry principal components.
Standardized Reporting and Quality Control: Implement genotype counts by case/control status, specify genetic models (additive/dominant/recessive), evaluate Hardy-Weinberg equilibrium, as well as imputation and quality control metrics, alongside prespecified analysis plans.
Pre-registered Systematic Review and Meta-analysis: Conduct a systematic review and meta-analysis based on the evidence compiled here, utilizing random-effects models and trans-ethnic meta-regression, inclusive of sensitivity analyses (e.g., omitting studies with deviations from Hardy-Weinberg equilibrium) and checks for small-study bias.
Functional Validation of Key Variants: Perform functional validation for significant variants (e.g., promoter assays for rs699947/rs833061; 3′UTR reporter assays for rs3025039; CRISPR editing in endothelial/pericyte models) and pursue integrative genomics studies.
Causal Inference and Translation: Explore Mendelian randomization to assess the relationship between VEGF levels and complications; develop and externally validate polygenic risk scores for diabetic retinopathy, diabetic nephropathy, diabetic peripheral neuropathy, and diabetic foot ulcers, using decision-curve analysis to evaluate clinical utility; and investigate pharmacogenetic signals that may influence responses to anti-VEGF treatments.
We have included a succinct "Future Directions" paragraph in the Discussion to reflect these points, which is highlighted in green (Page 36, line 1067 to 1078).

Reviewer 2 Report
Comments and Suggestions for Authors
Thanks to the authors for summarizing and updating the role of VEGF and their molecular mechanism in diabetic neuropathy and diabetic retinopathy at genetic level. Because VEGF role in disease pathogenesis and recovery remains controversial and this review will highlight the issues.
- Abbreviation EG-VEGF – line 52
- You mean VEGFR1 & VEGFR2 – line 52? Please correct it.
- Line 72-74; you mean VEGFA expression decreased in type-2 diabetes and this is the reason VEGF-VEGFR2 signaling is partially inhibited? Please clarify.
- Errera et al., 2007[51] Brazilian VEGFA rs2010963(-634G/C) PCR-RFLP 334 167 The CC genotype is an inde[1]pendent risk factor for the de[1]velopment of PDR in T2DM of European ancestry. Is this change not found in other population or there is no report available?
- “The VEGFA rs3025039 polymorphism appears to be more prominently associated 1057 with DR and PDR rather than T2DM “- Line 157. It means the rs3025039 polymorphism is not the primary factors of T2DM. Please explain.
- Table 5 is showing number of studies associated with different population. Is T2DM column showing “population has not shown any complications yet post T2DM’ and other groups has T2DM+ one of the complications?
- DPN, DR AND DNR is also a complication of type-1 diabetes. Just curious whether any study has included both type-1 & 2 to find an association in genetic polymorphism?
- The role of VEGF in macrovascular complications has been mostly reported deleterious. However, the report suggests that “individuals carrying the CC genotype had significantly higher serum VEGF levels, indicating a potential functional consequence of this promoting angiogenesis and vascular dysfunction”. On the other hand, T2DM shows less VEGF level and this is also a consequence of poor wound healing. Is there any clarity based on the literature reviews?
Author Response
# REVIEWER 2
Thanks to the authors for summarizing and updating the role of VEGF and their molecular mechanism in diabetic neuropathy and diabetic retinopathy at genetic level. Because VEGF role in disease pathogenesis and recovery remains controversial and this review will highlight the issues.
Answer: We appreciate the Reviewer’s positive evaluation of our manuscript. Your observations contributed significantly to improving the scientific clarity, precision, and robustness of the final version of our work. All suggestions were carefully considered and addressed either directly in the revised manuscript or in the specific responses below. The changes made to the text are clearly highlighted in green in the revised version. They are annotated with indications of the respective page and line numbers for transparency and ease of verification.
Abbreviation EG-VEGF – line 52
Answer: We appreciate the Reviewer’s comment. For enable a faster revision, the corrected sentence is highlighted in green (page 2, lines 51–52).
You mean VEGFR1 & VEGFR2 – line 52? Please correct it.
Answer: We appreciate the Reviewer’s comment. For enable a faster revision, the corrected sentence is highlighted in green (page 2, line 52).
Line 72-74; you mean VEGFA expression decreased in type-2 diabetes, and this is the reason VEGF-VEGFR2 signaling is partially inhibited? Please clarify.
Answer: Thank you for your question. We did not intend to suggest a uniform, global inhibition of VEGF–VEGFR2 signaling in T2DM. Instead, the pathway seems to be context-dependent. In various metabolic and vascular tissues, studies indicate a reduction in VEGFA expression accompanied by diminished VEGFR2 activation, including reduced phosphorylation, altered receptor internalization and trafficking, and weakened downstream signaling. Conversely, in the retinal microenvironment associated with diabetic retinopathy, VEGF levels are often elevated, which contributes to pathological angiogenesis. We have revised the text to better convey this nuance. All modifications are highlighted in green (page 2, lines 74–80).
Errera et al., 2007[51] Brazilian VEGFA rs2010963(-634G/C) PCR-RFLP 334 167 The CC genotype is an independent risk factor for the development of PDR in T2DM of European ancestry. Is this change not found in other population or there is no report available?
Answer: Thank you for raising this important point. The association described by Errera et al. (2007) in Brazilian patients of European ancestry is not unique. Similar associations between the C allele or CC genotype of rs2010963 and increased risk of DR/PDR were also reported in Indian cohorts (Suganthalakshmi 2006; Uthra 2008; Choudhury 2015), Egyptian cohorts (El-Shazly 2014; Kamal 2016), and Chinese cohorts (Yang 2010; Chen 2016). However, several studies in European, African, and Asian populations did not confirm this association (e.g., Buraczynska 2007; Petrovic 2008; Pirie 2014; Fan 2014). Moreover, some reports suggested even protective effects, such as the CG genotype in Chinese PDR patients (Jin 2021). We have clarified this variability in the revised text, which is highlighted in green (page 22, lines 453 to 457).
“The VEGFA rs3025039 polymorphism appears to be more prominently associated 1057 with DR and PDR rather than T2DM “- Line 157. It means the rs3025039 polymorphism is not the primary factors of T2DM. Please explain.
Answer: Thank you for the opportunity to clarify. Our statement reflects the findings of our scoping review’s descriptive mapping, which indicated that rs3025039 was reported more frequently in relation to DR/DPR than in the context of T2DM susceptibility. This does not imply that rs3025039 is not a contributor to T2DM; rather, the evidence pertaining to T2DM is heterogeneous. Furthermore, as this is a scoping review, we did not conduct quality appraisal or risk-of-bias assessments, nor did we perform a meta-analysis to formally compare effects. Accordingly, we have refined the wording in the manuscript to frame our findings as hypothesis-generating rather than confirmatory, and we acknowledge that a pre-registered systematic review and meta-analysis are planned to evaluate pooled associations.
We have clarified this aspect in the revised text, which is highlighted in green (page 36, lines 1085 - 1088).
Table 5 is showing number of studies associated with different population. Is T2DM column showing “population has not shown any complications yet post T2DM’ and other groups has T2DM+ one of the complications?
Answer: Thank you for your insightful question. We believe there may have been a wording mix-up, as the manuscript contains Figure 5 (not Table 5). In Figure 5, the "T2DM" column represents studies that assessed VEGF/VEGFR SNPs in patients with T2DM who had no documented microvascular or macrovascular complications at the time of evaluation. The other columns (such as DR, DN, DPN, DFU) indicate studies where T2DM patients were examined in the context of the respective complications. If a study reported on multiple endpoints (for instance, both T2DM-only and DR), it is counted in each applicable column. We have clarified this definition in the revised manuscript by improving the legend of Figure 5, which is highlighted in green (page 35, lines 1035–1037).
DPN, DR AND DNR is also a complication of type-1 diabetes. Just curious whether any study has included both type-1 & 2 to find an association in genetic polymorphism?
Answer: Thank you for your insightful question. We agreed that some complications of T2DM are also associated with T1DM. However, this scoping review was deliberately focused on T2DM, as outlined in our eligibility criteria (PCC framework). Consequently, studies focusing solely on T1DM or those with mixed T1DM/T2DM cohorts without T2DM-stratified results were excluded during the full-text screening process. We have clarified this scope in the Methods section and within the study-selection narrative. Mapping the evidence for T1DM is certainly of significant interest, and we intend to conduct a subsequent review that specifically addresses T1DM and its associated complications.
For clarity purposes, in the revised manuscript, we reinforced the PCC in the Methods to make this scope explicit, which is highlighted in green (page 6, lines 178–185).
The role of VEGF in macrovascular complications has been mostly reported deleterious. However, the report suggests that “individuals carrying the CC genotype had significantly higher serum VEGF levels, indicating a potential functional consequence of this promoting angiogenesis and vascular dysfunction”. On the other hand, T2DM shows less VEGF level, and this is also a consequence of poor wound healing. Is there any clarity based on the literature reviews?
Answer: Thank you for raising this important point. The observations you mentioned are not mutually exclusive but rather reflect the compartment- and tissue-specific biology of VEGF in diabetes. In certain ocular and macrovascular contexts, elevated levels of VEGF (such as in the vitreous or within atherosclerotic plaques) are associated with pathological angiogenesis, increased permeability, and plaque neovascularization, which can have detrimental effects. Consistently, the rs2010963 CC genotype and the rs2305948 CC genotype have been linked to higher VEGF levels in some cohorts, potentially amplifying these local angiogenic processes.
In contrast, in the context of cutaneous wounds in individuals with T2DM, chronic hyperglycemia and inflammatory stress can attenuate HIF-1α–VEGFA induction and VEGFR2 signaling (including receptor phosphorylation, internalization, and downstream pathways), resulting in impaired angiogenesis and poor wound healing. Additionally, circulating VEGF (found in serum or plasma) serves as an imperfect proxy for tissue signaling, as it can be influenced by factors such as platelet release, soluble receptors like sFLT1, and anti-angiogenic isoforms. Therefore, elevated serum VEGF or genotype-related increases do not necessarily imply enhanced local VEGF activity in wounds.
We have revised the text to clarify this context dependence and have softened the language to avoid suggesting a uniform global inhibition or activation. As a scoping review, our statements are intended to be hypothesis-generating, and we did not conduct a meta-analysis. All modifications are highlighted in green (page 31, lines 880–886).

Reviewer 3 Report
Comments and Suggestions for Authors
The manuscript provides a comprehensive scoping review examining the association between VEGF gene polymorphisms and T2DM-related complications.
The conclusion appropriately emphasises potential biomarker roles but could be more explicit about the clinical implications or how these findings could influence management strategies.
Author Response
#REVIEWER 3
The manuscript provides a comprehensive scoping review examining the association between VEGF gene polymorphisms and T2DM-related complications.
Answer: We sincerely thank the Reviewer for the favorable evaluation of our study. We also appreciate the valuable suggestion that several points need further clarification. Below, we provide detailed, point-by-point response and have revised the manuscript accordingly to improve clarity and rigor. All modifications are highlighted in green in the revised version.
The conclusion appropriately emphasizes potential biomarker roles but could be more explicit about the clinical implications or how these findings could influence management strategies.
Answer: Thank you for this valuable suggestion. We have expanded the Conclusion to make the clinical implications more explicit while maintaining appropriate caution for a scoping review. All modifications are highlighted in green in the revised version (page 36, lines 1099–1106).
From a clinical perspective, these polymorphisms may serve as potential biomarkers to identify individuals at higher risk for developing microvascular or macrovascular complications, allowing for earlier surveillance and targeted interventions. Furthermore, insights into VEGF-related pathways may inform the development of novel therapeutic strategies, including precision medicine approaches aimed at modulating angiogenesis in susceptible individuals. Integrating genetic information into risk assessment models could ultimately improve individualized management and reduce the burden of diabetes-related complications.
